# Cytosolic sorting platform complexes shuttle type III secretion system effectors to the injectisome in *Yersinia enterocolitica*

Stephan Wimmi ●[1,7], Alexander Balinovic[2,3,4,6,7], Corentin Brianceau ●[1,7], Katherine Pintor ●[1,7], Jan Vielhauer[1], Bartosz Turkowyd[2,3,4,6], Carlos Helbig ●[1], Moritz Fleck[1], Katja Langenfeld[1], Jörg Kahnt[1,5], Timo Glatter ●[5], Ulrike Endesfelder ●[2,3,4,6,8] ✉ & Andreas Diepold ●[1,3,8] ✉

Bacteria use type III secretion injectisomes to inject effector proteins into eukaryotic target cells. Recruitment of effectors to the machinery and the resulting export hierarchy involve the sorting platform. These conserved proteins form pod structures at the cytosolic interface of the injectisome but are also mobile in the cytosol. Photoactivated localization microscopy in *Yersinia enterocolitica* revealed a direct interaction of the sorting platform proteins SctQ and SctL with effectors in the cytosol of live bacteria. These proteins form larger cytosolic protein complexes involving the ATPase SctN and the membrane connector SctK. The mobility and composition of these mobile pod structures are modulated in the presence of effectors and their chaperones, and upon initiation of secretion, which also increases the number of injectisomes from ~5 to ~18 per bacterium. Our quantitative data support an effector shuttling mechanism, in which sorting platform proteins bind to effectors in the cytosol and deliver the cargo to the export gate at the membrane-bound injectisome.

Bacteria have evolved a variety of secretion systems to transport proteins across the bacterial cell wall and even directly into target cells[1,2]. The type III secretion system (T3SS) is the basis for both export of flagellar subunits and the injection of proteins from the bacterial cytosol into the cytoplasm of eukaryotic target cells by injectisomes[3,4]. Hereafter, we use 'T3SS' to refer to the injectisome, which is essential for the virulence of important animal and plant pathogens but also participates in commensalism and symbiosis. While this functional diversity is reflected in a wide range of species-specific translocated effector proteins, the components[5,6] and overall structure[7–9] of the injectisome itself are conserved (Fig. 1a). After the assembly of the core of the machinery[10–12], the  system exports early secretion substrates such as the needle subunit. Upon needle assembly, middle substrates, such as the protein-forming needle tip, are exported and assembled. At this point, the injectisome is in a steady state until host cell contact or until chemical cues trigger the export of its late substrates, the virulence effectors. Effectors are injected into target cells in a one-step process, which is facilitated by a hydrophobic translocon pore in the target membrane[13–15].

Although the translocation of effectors from the bacterial cytosol into host cells is the central feature of virulence-associated type III secretion, the recruitment and path of the effectors to and through the injectisome are poorly understood: How are the effectors recruited

[1]Department of Ecophysiology, Max Planck Institute for Terrestrial Microbiology, Marburg, Germany. [2]Department of Systems and Synthetic Microbiology, Max Planck Institute for Terrestrial Microbiology, Marburg, Germany. [3]SYNMIKRO, Center for Synthetic Microbiology, Marburg, Germany. [4]Department of Physics, Carnegie Mellon University, Pittsburgh, PA, USA. [5]Mass Spectrometry and Proteomics Facility, Max Planck Institute for Terrestrial Microbiology, Marburg, Germany. [6]Present address: Institute for Microbiology and Biotechnology, Rheinische Friedrich-Wilhelms-Universität Bonn, Bonn, Germany. [7]These authors contributed equally to this work: Stephan Wimmi, Alexander Balinovic, Corentin Brianceau, Katherine Pintor. [8]These authors jointly supervised this work: Ulrike Endesfelder, Andreas Diepold. ✉e-mail: endesfelder@uni-bonn.de; andreas.diepold@mpi-marburg.mpg.de

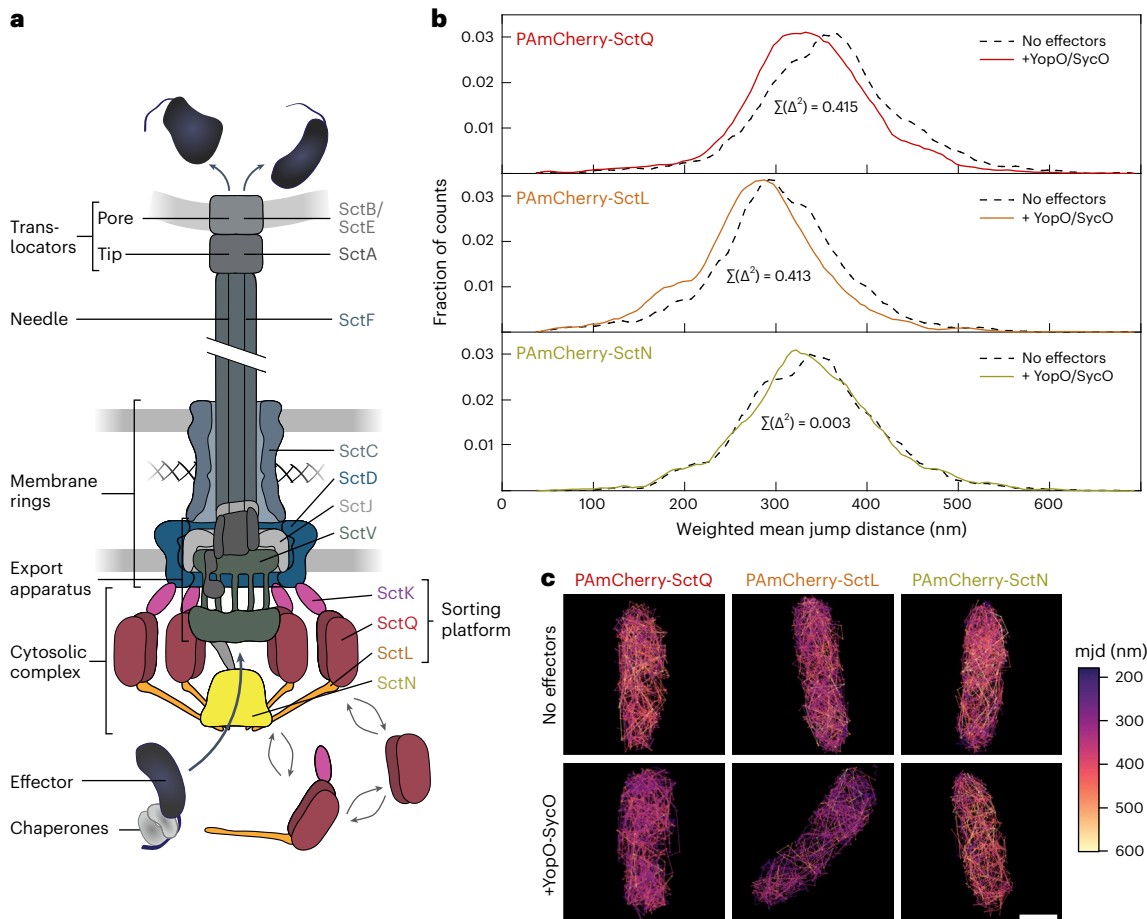

**Fig. 1 | Effectors directly interact with sorting platform components in live bacteria. a**, Schematic representation of effector export by the T3SS. Proteins studied in this report are highlighted. **b**, Mobility of indicated sorting platform components in *Y. enterocolitica* lacking all other components of the T3S (pYV⁻) in the presence (coloured lines) and absence (dashed black lines) of the T3SS effector YopO and its chaperone SycO. Histogram of mean jump distances (mjd) of molecular diffusion, weighted for the number of jump distances. The sums of

squared differences ($\Sigma(\Delta^2)$) of the cumulative distributions were 0.415 for SctQ, 0.413 for SctL and 0.003 for SctN; the squared correlation coefficients ($r^2$) of the distributions were 0.906 for SctQ, 0.906 for SctL and 0.984 for SctN. **c**, Trajectories in representative bacteria. Scale bar, 0.5 μm. Numbers of trajectories and replications for single-particle tracking experiments are summarized in Supplementary Table 2.

from the bacterial cytosol? Which machinery components do they initially bind to, and do the interactions between effectors and T3SS components change upon the initiation of protein secretion, for example, by host cell contact?

The 'sorting platform', a set of proteins consisting of the cytosolic T3SS components SctQ, SctK and SctL, may govern these events (Fig. 1a). SctQ is a homologue of the flagellar C-ring proteins FliM and FliN. While full-length SctQ is mostly homologous to FliM, an internal translation initiation site leads to the expression of an additional C-terminal fragment, SctQ$_C$, with strong homology to FliN[16,17]. Surprisingly, the quaternary structure of the cytosolic complex is strikingly different between the flagellum and the injectisome. While in the flagellum, FliMN forms a continuous ring structure involved in switching the direction of flagellar rotation[18], six distinct pod structures form the cytosolic interface of the injectisome[7–9,19,20]. Per pod, one copy of SctK, a protein without clear flagellar homologue[8,21,22], connects a core of several copies of SctQ and probably SctQ$_C$ to the inner membrane (IM) ring. A dimer of SctL links SctQ to the ATPase SctN, which forms a central hexamer connecting all six pods (Fig. 1a). SctN may prepare effectors for export by detaching their chaperones[23]. The SctQ, SctK and SctL of the *Salmonella* pathogenicity island 1 (SPI-1) T3SS have been found to form high molecular weight complexes that co-migrate with selected T3SS substrates in native gel electrophoresis[24]. Notably,

an effector protein, a late T3SS export substrate, was less abundant in these complexes in the presence of a translocator protein, which must be secreted earlier to form the pore in the host cell (Fig. 1a). This indicates that export order may depend on the affinity of substrates to SctQKL, which is the basis of the 'sorting platform' model[24]. The direct binding of a general effector chaperone to SctQ has also been shown in *Chlamydia* by yeast-two-hybrid analysis[25]; furthermore, pull-down experiments indicate a direct interaction of SctQ with T3SS effectors in *Shigella flexneri*[26].

The sorting platform components and SctN are not stably bound to the injectisome. Besides their injectisome-bound state, they are also present as soluble proteins and subcomplexes in the bacterial cytosol[24,25,27–34]. Notably, sorting platform subunits exchange between the soluble and the injectisome-bound state[35]. This exchange is correlated with the function of the injectisome[28,35,36]. We recently showed that the dynamic nature of the sorting platform allows for the adaptation of protein secretion to external conditions[37]. Exchange of the sorting platform component SctQ is also the basis of a recently developed light-controlled T3SS, based on optogenetic membrane sequestration of the cytosolic SctQ, which prevents secretion[38].

In line with its continuous binding and unbinding at the injectisome, the sorting platform has been proposed to shuttle effectors from the cytosol to the injectisome or increase the local concentration

of export substrates at the injectisome[5,33,35,39,40]. This hypothesis could not be tested in the absence of quantitative data on the interaction between effectors and the sorting platform. Thus, it has remained unclear whether the exchange of sorting platform subunits is an integral part of the export process itself.

Despite their presumed key role in effector selection and export, interactions between the sorting platform and the effectors have rarely been described in protein interaction studies, whether in vitro or in vivo. This lack of detection might be a consequence of the inherently transient nature of such interactions during the export process. We used complementary in vivo approaches, proximity labelling and single-particle tracking, to circumvent this problem. Proximity labelling clearly indicated interactions of the central sorting platform component SctQ with T3SS effectors in live bacteria. To further characterize these interactions, we measured the location and mobility of the sorting platform proteins in live *Yersinia enterocolitica* by single-particle tracking super-resolution microscopy. The large number of well-characterized functional fluorescent fusion proteins[10,28,41] and the possibility to efficiently control secretion by modulating external calcium levels[42] make *Y. enterocolitica* an excellent model organism for such studies. In agreement with previous studies, we detected that the majority of sorting platform components were not bound to injectisomes. Our results provide strong evidence for a direct interaction between effectors and the sorting platform components SctQ and SctL in live bacteria. This interaction occurs in the cytosol and was observed even in the absence of essential membrane components of the T3SS. Our results provide new insights into the crucial initial steps of type III secretion and the path of T3SS effectors before their translocation into target cells, and could serve as a stepping stone for further investigation of other secretion systems and their targeted inhibition.

## Results

### SctQ and SctL directly bind to effectors in live bacteria

To analyse the interactions between sorting platform components and export substrates, we first performed co-immunoprecipitation experiments in *Y. enterocolitica*. To this aim, we replaced the genes for the sorting platform components SctQ and SctL, as well as the interacting ATPase SctN, with N-terminal PAmCherry fusions by allelic exchange[43]. The fusion proteins were thus expressed from their native genetic locus. All fusion proteins were stable and fully functional for secretion (Supplementary Fig. 1). Bacteria were incubated at 37 °C, which induces expression and assembly of the T3SS components, under non-secreting or secreting conditions. After collection and lysis, we performed immunoprecipitation with magnetic beads targeting mCherry (Supplementary Fig. 2). Interacting proteins were identified by shotgun liquid chromatography–mass spectrometry (LC–MS)-based proteomics. The results were in line with the interaction network within the cytosolic components found in earlier studies[26–28,31,44]: SctL was highly significantly enriched in the PAmCherry-SctQ immunoprecipitation, while the significance of enrichment of the peripheral components SctK and especially SctN was lower (Extended Data Fig. 1). Although some export substrates were detected, their enrichment was less consistent, probably due to their transient interaction with the sorting platform. Interpretation of these results is complicated by unspecific interactions of the sorting platform proteins, visible in the asymmetric profiles of the volcano plots. Specific interactions between purified SctQ (in complex with SctQ_C) and the native *Y. enterocolitica* effector YopO (in complex with its chaperone SycO) were detected upon crosslinking in vitro, but not in biolayer interferometry (Extended Data Fig. 2). All these results are compatible with a transient interaction of the sorting platform and T3SS export cargo.

As such interactions are best analysed in live bacteria, we performed in vivo proximity labelling[45] using an endogenously expressed fusion of the unspecific biotin ligase miniTurbo[46] to SctQ. Proteins near miniTurbo-SctQ are preferentially biotinylated and can then

**Table 1 | Proximity labelling indicates the interaction of the sorting platform protein SctQ with T3SS effectors in situ**

| Protein | Enrichment factor | *P* value |
|---|---|---|
| Sorting platform protein SctQ (bait) | 16.61 | 0 |
| Effector YopH | 2.41 | 0.022 |
| Effector YopT | 2.22 | 0.005 |
| Effector YopO | 2.02 | 0.023 |
| Sorting platform protein SctL | 1.93 | 0.004 |
| Effector/gatekeeper SctW/YopN | 1.86 | 0.060 |
| Effector YopM | 1.66 | 0.063 |
| Effector YopP | 1.61 | 0.104 |
| Effector YopR | 1.57 | 0.029 |

List of all T3SS proteins enriched by a factor of >1.5 after streptavidin purification of in situ proximity biotinylated proteins in a strain natively expressing miniTurbo-SctQ, compared to a miniTurbo control expressed at a comparable level. *P* values were calculated by a two-sided *t*-test as implemented in Perseus[88]. See Supplementary Table 1 for details.

be purified and analysed by proteomics. Strikingly, the most highly enriched proteins not only included SctQ and its known interaction partner SctL, but also six of the eight *Yersinia* T3SS effectors (Table 1). This enrichment was highly specific, as no other T3SS component was similarly enriched (Supplementary Table 1).

To better characterize the interactions between the sorting platform and effectors, we next performed single-particle tracking photoactivated localization microscopy (sptPALM)[47] in live bacteria. This method allows tracking of the movement of labelled proteins by high-speed microscopy. This movement is expressed as the mean jump distance (mjd) of the protein between successive microscopy images. Higher mjd values indicate faster diffusion and a smaller size of the protein or protein complex (Supplementary Text 1). As an in situ method, sptPALM can be used to study transient and stable interactions in live bacteria. To specifically analyse the direct binding of effectors to single sorting platform components, we first systematically tested those interactions in *Y. enterocolitica* strains lacking the plasmid for *Yersinia* virulence (pYV), which encodes all T3SS components. In this pYV⁻ background, we co-expressed from plasmids (1) the N-terminal PAmCherry fusions to SctQ, SctL or SctN used in the previous experiment and (2) the large native effector YopO and its chaperone SycO. In bacteria with a functional T3SS, the plasmid used for the expression of YopO and SycO allowed for the secretion of YopO, indicating that YopO and SycO are expressed and functionally interact (Supplementary Fig. 3). We then compared the mobility of the sorting platform proteins in the presence and absence of YopO and SycO by sptPALM. PAmCherry alone, which we expressed in the same strain background as a control, displayed the fastest diffusion, as expected due to its lower molecular weight (Supplementary Fig. 4 and Data 1). Our experiments show that in live bacteria, diffusion of both PAmCherry-SctQ and PAmCherry-SctL was slowed down in the presence of YopO and SycO, revealing an influence, most probably binding, of the chaperone–effector pair (Fig. 1b,c and Supplementary Data 1). In contrast, diffusion of the ATPase PAmCherry-SctN was not significantly affected by the presence of the effector and its chaperone (Fig. 1b,c and Supplementary Data 1).

These results indicate a direct binding of effectors to SctQ and SctL, which we next analysed under native conditions in live wild-type (WT) *Y. enterocolitica* expressing all components of the T3SS.

### Effectors influence mobility of sorting platform complexes

To study the binding of effectors to the sorting platform proteins and the effect of secretion activation in live bacteria expressing functional T3SS, we focused on SctQ, on the basis of the strong and well-defined

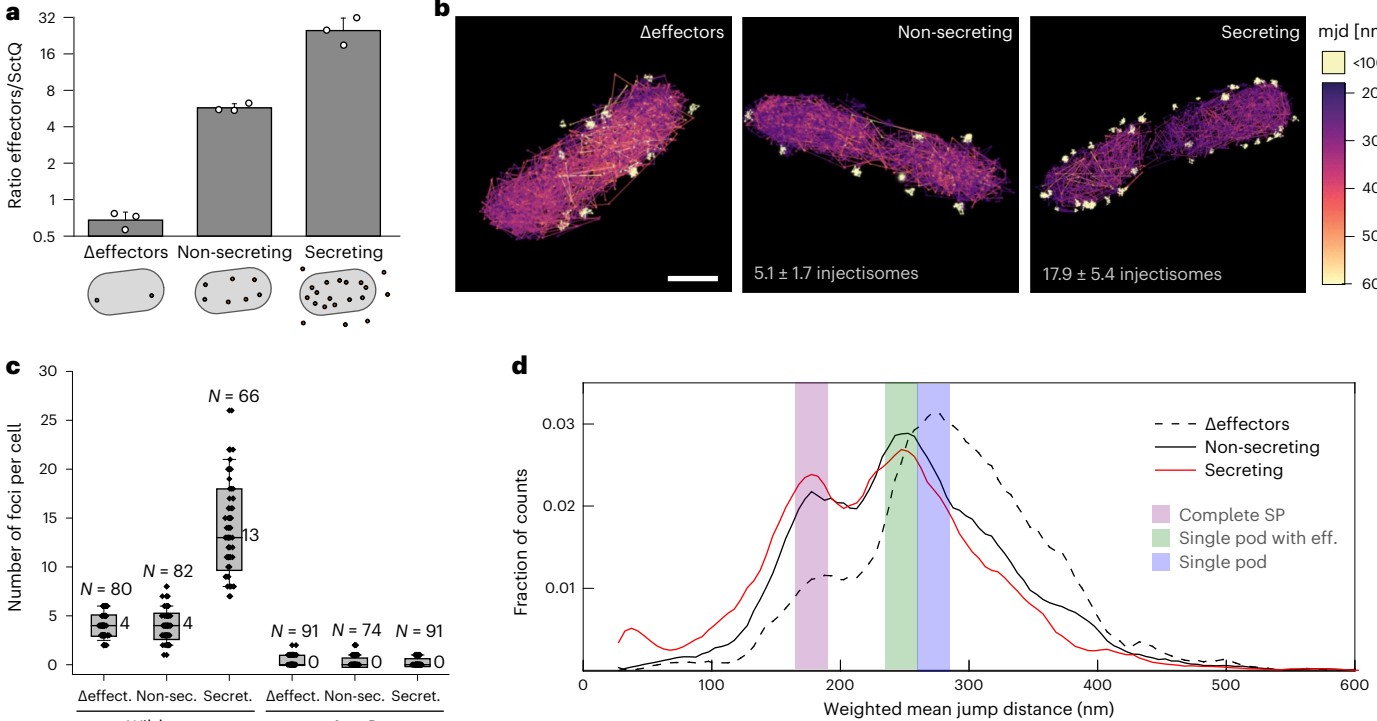

**Fig. 2 | Presence of effectors influences the localization and mobility of the main sorting platform component SctQ. a**, Ratio of the total quantity of T3SS effectors and SctQ as detected by label-free total cell mass spectrometry in the indicated *Y. enterocolitica* strain backgrounds and conditions. *n* = 3 biologically independent samples, single data points indicated, error bars denote s.d. See Supplementary Table 3 for underlying proteomics data. **b**, Representative PAmCherry-SctQ sptPALM trajectories, colored according to their mjd value. Immobile trajectories (representing SctQ in sorting platforms bound to the membrane-bound injectisome complex) are colored white for better visibility— hence independent of the quantitative color scale. Scale bar, 0.5 µm; average number of injectisomes indicated at the bottom. **c**, Quantification of stable PAmCherry-SctQ foci per bacterial cell detected by sptPALM in the indicated

*Y. enterocolitica* strain backgrounds and conditions. Numbers next to each bar plot indicate the corresponding median value; total numbers of analysed bacterial cells are indicated on top of each bar plot. Box denotes mean and s.d.; whisker range corresponds to 5–95%. **d**, Mobility of cytosolic PAmCherry-SctQ molecules in the indicated strain background and conditions. Histogram of mjd of molecular diffusion, weighted by the number of jump distances. Vertical bars indicate the range of maximal mean jump distances ±12.5 nm, interpreted as 'single pod' (260–285 nm, blue), 'single pod with effectors' (235–260 nm, green) and 'complete sorting platform' (170–195 nm, purple). Numbers of trajectories and replications for single-particle tracking experiments are summarized in Supplementary Table 2.

interaction pattern of PAmCherry-SctQ with YopO/SycO shown in the previous experiment.

We analysed the mobility of PAmCherry-SctQ, expressed from the native SctQ promoter, in live bacteria in three different strain backgrounds and conditions: (1) a strain lacking the main effectors YopH,O,P,E,M,T[48] (hereafter called Δeffectors), (2) the *Y. enterocolitica* wild-type strain MRS40 expressing all effectors and (3) the same strain under secreting conditions. As expected, the ratio of effectors per SctQ increased sequentially from (1) to (3) (Fig. 2a and Supplementary Table 3).

Analysis of PAmCherry-SctQ mobility by sptPALM revealed the presence of two distinct states of proteins: while one subset of PAmCherry-SctQ was stably localized in foci at the bacterial membrane, a second, mobile subset was distributed throughout the cytosol (Fig. 2b). The stable foci are injectisome-bound PAmCherry-SctQ. In contrast, the diffuse population represents a cytosolic pool of unbound PAmCherry-SctQ protein and subcomplexes, a heterogeneous population leading to a broad distribution of mobility (Supplementary Text 2). A similar distribution had been observed earlier for SctQ and SctL in *Y. enterocolitica*, as well as the sorting platform proteins of the SPI-1 T3SS[29,30,35]. We found an average of 3.9 ± 1.3 (median = 4) injectisomes under non-secreting conditions and 13.8 ± 4.2 (median = 13) injectisomes under secreting conditions in live *Y. enterocolitica* (Fig. 2c). This is in line with the known upregulation of assembly and resulting

higher number of injectisomes in secreting bacteria[49]. Almost no injectisomes were detected in the absence of SctD (median = 0; Fig. 2c) and in the cytosol (median = 0–1; Supplementary Fig. 6), highlighting the specificity of the detection. Considering the incomplete detection of PAmCherry-SctQ molecules via sptPALM imaging (Supplementary Fig. 7 and Text 3), we determined the underlying absolute number of injectisomes to be 5.1 ± 1.7 injectisomes per non-secreting bacterium and 17.9 ± 5.4 injectisomes per secreting bacterium. As noticed in earlier studies[9,49], many injectisomes were arranged in clusters, especially under secreting conditions (Fig. 2b right), indicative of an integration of new injectisomes close to existing ones.

Similar to the majority of the SctQ molecules (Fig. 2b), effectors and chaperones localize throughout the cytosol in live wild-type *Y. enterocolitica* (ref. 50 and Supplementary Fig. 8). If the effectors bind to SctQ in the cytosol in these native conditions, an increasing ratio of effectors to sorting platform components should slow down SctQ. To exclude any confounding influence of injectisome-bound components, we limited our sptPALM analysis to cytosolic trajectories, which represent ~85% of all trajectories (Supplementary Fig. 9). In the absence of effectors, PAmCherry-SctQ mobility peaks at an mjd of 272.5 nm (mean of 5 nm, bin 270–275 nm) (Fig. 2d), significantly slower than in the pYV⁻ strain lacking all other T3SS components (Fig. 1b,c). This behaviour indicates the formation of larger complexes that include other T3SS components. We defined this fraction (with an

**Table 2 | Predicted mobility values for the mean jump distance of sorting platform complexes in the indicated strain backgrounds based on their molecular compositions**

| Wild-type strain PAmCh-SctQ | Stoichiometry (copy numbers) of possible complexes | | | | | | Without effector(s) | | | With effector(s) | | |
|---|---|---|---|---|---|---|---|---|---|---|---|---|
| | SctK | SctQ | SctQ$_C$ | SctL | SctN | Eff. chap. compl. | MW (kDa) complex | Calculated mjd (nm) | Apparent diffusion coefficient (μm²s⁻¹) | MW (kDa) complex | Calculated mjd (nm) | Apparent diffusion coefficient (μm²s⁻¹) |
| Single pod (*a*) | 1 | 4 | 8 | 2 | 1 | 4 | 451.5 | **272.5** | 2.34 | 714.6 | 252.4 | 2.00 |
| Single pod lacking SctK (*b*) | | 4 | 8 | 2 | 1 | 4 | 427.5 | 275.0 | 2.38 | 690.6 | 253.9 | 2.02 |
| Single pod lacking SctN (*c*) | 1 | 4 | 8 | 2 | | 4 | 403.8 | 277.6 | 2.43 | 666.9 | 255.4 | 2.05 |
| SctK(Q(Q$_C$)$_2$)$_4$ (*d*) | 1 | 4 | 8 | | | 4 | 353.9 | 283.8 | 2.54 | 617.0 | 258.7 | 2.10 |
| (SctQ(Q$_C$)$_2$)$_4$ (*e*) | | 4 | 8 | | | 4 | 330.0 | 287.1 | 2.60 | 593.0 | 260.4 | 2.13 |
| SctQ(Q$_C$)$_2$ (*f*) | | 1 | 2 | | | 1 | 82.5 | 361.7 | 4.14 | 148.3 | 328.1 | 3.40 |
| Complete sorting platform (*g*) | 6 | 24 | 48 | 12 | 6 | 24 | 2,708.9 | 202.2 | 1.27 | 4,287.4 | 187.3 | 1.09 |
| **pYV⁻ strains** | | | | | | | | | | | | |
| PAmCh-SctQ | | 1 | | | | 1 | 62.5 | 378.9 | 4.54 | 179.8 | 317.7 | 3.19 |
| PAmCh-SctQ(Q$_C$)$_2$ | | 1 | 2 | | | 1 | 82.5 | 361.7 | 4.14 | 199.8 | 312.2 | 3.07 |
| **Add. poss. pod compositions** | | | | | | | | | | | | |
| Sct(Q(Q$_C$)$_2$)$_2$L$_2$N | | 2 | 4 | 2 | 1 | 2 | 262.6 | 298.3 | 2.83 | 497.2 | 268.2 | 2.29 |
| Sct(Q(Q$_C$)$_2$)$_2$L$_2$ | | 2 | 4 | 2 | | 1 | 214.8 | 308.4 | 3.03 | 332.1 | 286.8 | 2.62 |
| SctQ(Q$_C$)$_2$L$_2$N | | 1 | 2 | 2 | 1 | 1 | 180.1 | 317.6 | 3.21 | 297.4 | 292.1 | 2.72 |
| SctKQL$_2$N | 1 | 1 | | 2 | 1 | 1 | 184.0 | 316.5 | 3.19 | 301.3 | 291.5 | 2.70 |

Calculations are based on the Stokes–Einstein relation and the measured base value of 272.5 nm mean jump distance for the SctK(Q(Q$_C$)$_2$)$_4$L$_2$N complex designated as 'single pod' (see main text and Supplementary Text 1 for details). Letters in italics in the left column refer to depictions of the respective complexes in Extended Data Fig. 4. While single-molecule counting experiments (Extended Data Fig. 3) indicate a 2:1 stoichiometry of SctQ:SctL, other studies proposed different stoichiometries of the pod structure[31,89]. Some of these stoichiometries are included at the bottom of the table; the general relation of molecular weight and predicted mobility is described in Supplementary Text 1. Bold font denotes the measured value of the main experimental mjd peak in the experiments (Fig. 2d), which was used as base value for all calculations; eff. chap. compl. refer to effector–chaperone complexes. MW, molecular weight. For conversion of mjd to apparent diffusion coefficient, see Supplementary Text 4.

mjd of 260–285 nm) as a 'single pod'. On the basis of this finding and interaction data from both our group and others[27,28,31,51], we tentatively assigned this peak to an SctK(Q(Q$_C$)$_2$)$_4$L$_2$N subcomplex (Table 2). This composition of the pod structure is further supported by the 2:1 ratio of SctQ:SctL bound at the injectisome, which we quantified by molecule-counting experiments in live and fixed cells (Extended Data Fig. 3). In the presence of effectors, the primary observed SctQ mobility peak shifts to 252.5 nm. In addition, a pronounced fraction of proteins was slowed down even further, to a peak mjd of 177.5 nm (Fig. 2d). Since this effect was introduced by the addition of the effectors, we reasoned that these peaks might correspond to 'single pods with effectors' (235–260 nm) and 'complete cytosolic sorting platforms', corresponding to a complex of six pod structures (165–190 nm). Using an mjd of 272.5 nm for the SctK(Q(Q$_C$)$_2$)$_4$L$_2$N complex as a reference, we could predict the expected diffusion of these and other substructures, on the basis of the known molecular mass of the individual components (Table 2, Extended Data Fig. 4 and Supplementary Text 1). Importantly, we could also calculate the predicted impact of binding of an average effector–chaperone complex to the sorting platform substructures. The predicted mobility of 252.4 nm for 'single pods with effectors' and 187.3 nm for 'complete cytosolic sorting platforms with effectors' upon binding of one effector–chaperone complex per SctQ are in striking agreement with the respective observed mobility peaks (Fig. 2d, and Tables 2 and 3). Activation of secretion retains both the 'single pods with effectors' and the 'complete cytosolic sorting platforms with effectors' peaks, but slightly shifts the equilibrium towards the latter (Fig. 2d). Free diffusing PAmCherry, used as a control, showed a fast diffusion with a similar broad mjd distribution in both secreting and non-secreting conditions (Supplementary Fig. 10). This indicates that parameters that influence overall protein mobility, such as the viscosity and molecular crowding

**Table 3 | Measured peak mean jump distance values**

| PAmCh-SctQ | Conditions | Main peak | | Secondary peak | |
|---|---|---|---|---|---|
| | | Observed mjd value (nm) | Apparent diffusion coefficient (μm²s⁻¹) | Observed mjd value (nm) | Apparent diffusion coefficient (μm²s⁻¹) |
| WT | Δeffectors | **272.5** | 2.34 | 187.5 | 1.09 |
| | non-secreting | 252.5 | 2.00 | 177.5 | 0.98 |
| | secreting | 247.5 | 1.92 | 177.5 | 0.98 |
| ΔsctD | Δeffectors | 262.5 | 2.17 | | |
| | non-secreting | 242.5 | 1.84 | | |
| ΔsctK | Δeffectors | 272.5 | 2.34 | | |
| | non-secreting | 242.5 | 1.84 | | |
| ΔsctL | Δeffectors | 302.5 | 2.89 | | |
| | non-secreting | 292.5 | 2.70 | | |
| pYV⁻ | | 362.5 | 4.16 | | |
| | +YopO-SycO$_2$ | 322.5 | 3.28 | | |

Peak mjd values for the indicated strains and conditions, as displayed in Figs. 1b, 2d and 3. Bold font denotes the measured value of the main experimental mjd peak (Fig. 2d) that serves as a reference value to predict the diffusion of other substructures. Mjd values given as the centre of the respective 5 nm bin. For conversion of mjd to apparent diffusion coefficient, see Supplementary Text 4.

in the cytosol of *Y. enterocolitica*, are comparable under both conditions and similar to the strain lacking the complete T3SS (pYV⁻; Fig. 1b,c). Notably, using the 'single pod' mjd peak value of 272.5 nm as a basis, we could re-evaluate the data obtained for the direct protein interactions in the initial experiment (Fig. 1b,c). The observed peak for

PAmCherry-SctQ in the pYV⁻ strain background closely matches the predicted mobility of a PAmCherry-SctQ($Q_C$)$_2$ complex, which probably constitutes the basic building block of the sorting platform[17,52], both alone (observed peak mjd of 362.5 nm, predicted 362 nm) and with the bound YopO-SycO$_2$ effector–chaperone complex (observed peak mjd of 322.5 nm, predicted 312 nm) (Tables 2 and 3).

Taken together, these data imply that effectors directly bind to SctQ in the bacterial cytosol. SctQ itself is present mainly as part of larger sorting platform complexes whose composition is further changed by the activation of secretion.

### Effector binding occurs in the cytosol in wild-type bacteria

To investigate whether the initial interaction between SctQ and T3SS effectors occurs in the cytosol, we performed sptPALM of PAmCherry-SctQ in a strain lacking the IM ring component SctD. In this strain, the sorting platform proteins are completely cytosolic[10,28]. The mobility distribution in the *ΔsctD* background, including the shift from the 'pod' to the 'pod with effectors' peak (Fig. 3 top), was strikingly similar to that in the WT strain (Fig. 2d). The main difference was the expected lack of sorting platform complexes at the membrane and a reduced prominence of the 'complete sorting platform' peak. In contrast to the wild-type strain, the activation of secretion did not lead to major changes of diffusion characteristics in *ΔsctD*, which was also expected, as secretion and the upregulation of effector expression do not take place in the absence of SctD.

### SctK and SctL are not required for effector binding

Having established that effector binding to SctQ occurs in the cytosol independently of the basal body in wild-type strains, we wanted to determine the influence of the other sorting platform components SctK and SctL. To this aim, we tracked the diffusion of PAmCherry-SctQ in *ΔsctK* and *ΔsctL* strains. Similar to the *ΔsctD* strain, sorting platform proteins are exclusively cytosolic in these strains[10,28]. Absence of SctK led to a similar but slightly faster diffusion compared with *ΔsctD* (Fig. 3 centre), compatible with the lack of one SctK molecule from the sorting platform pods. Notably, the presence of effectors still caused a clear reduction in diffusion, with peak values compatible to the respective expected 'pod' and 'pod with effector' values (Tables 2 and 3). These results indicate that while SctK forms part of the pod structure, it is not essential for export substrate binding to SctQ, which is consistent with our measurements in the pYV⁻ strains (Fig. 1b,c).

The deletion of *sctL* had a significant effect on the diffusion profile of PAmCherry-SctQ. The 'complete sorting platform' peak was absent, in line with the central location of SctL in this complex. The presence of effectors clearly influenced the diffusion of SctQ even in the absence of SctL (Fig. 3 bottom), again in agreement with the previous finding that SctQ directly binds to effectors. The slower 'pod with effectors' peak was reduced to a shoulder and shifted to faster diffusion (~260–265 nm), in line with the smaller pod size in the absence of SctL (predicted mjd = 259 nm; Tables 2 and 3). However, the fraction of faster-diffusing SctQ (peak mjd at 292.5 nm) was increased. This peak at slightly below 300 nm was also one of the main peaks in the absence of effectors, indicating that the presence of SctL stabilizes effector binding. In the absence of effectors, another population (peak mjd of 337.5 nm) was prominent, which might correspond to an SctK-Q-$Q_{C,2}$ complex (predicted mjd = 347 nm).

Overall, the diffusion profile of PAmCherry-SctQ *ΔsctL* closely resembled that of PAmCherry-SctQ in pYV⁻ (Fig. 1b,c), indicating that SctL is not essential for interaction of effectors with SctQ, but stabilizes effector binding and allows for the formation of higher-order structures.

## Discussion

In recent years, structural and functional studies have improved our understanding of the export mechanism of the T3SS. Cryo-electron microscopy of the IM export apparatus indicated possible paths of the exported proteins[53,54]. Further experiments revealed specific interactions between the N-terminal export signal of export substrates and export apparatus proteins[55–59]. We also gained important insight into the structure and dynamics of the cytosolic components of the T3SS, also called the sorting platform. On the one hand, the sorting platform constitutes a clearly defined structure that can be visualized in situ[7–9,19,60], but on the other hand, it is dynamic: The components also form soluble cytosolic complexes[27–31,61], and there is an exchange between cytosolic and injectisome-bound proteins[35]. This exchange may explain the lack of sorting platforms in purified T3SS needle complexes. In addition, this feature has recently been exploited to control T3SS function by optogenetics[38].

However, the physiological relevance of this exchange has remained unknown; in fact, it is not even clear why the sorting platform proteins are necessary for protein export through the T3SS in the first place. Protein interactions in transport processes are by definition transient and thus difficult to detect; studies on these are accordingly scarce[24–26]. In *Salmonella* SPI-1, binding of a virulence effector to a high molecular weight SctKQL complex is reduced in the presence of a translocator, which needs to be exported earlier. This suggestive observation is the basis of the sorting platform model, according to which the affinity of different export substrates for the SctQKL complex determines their export order[24].

Although the sorting platform model has gained popularity, there is very limited amount of additional supporting data, and it was unclear whether it can be generalized to other organisms. In particular, interactions between effectors and the sorting platform had never been visualized before in live bacteria and it was unknown where such interactions occurred. In this work, we explicitly focused on this challenge. 'Classical' biochemical pull-down interaction experiments confirmed known interactions within the sorting platform (Supplementary Figs. 2 and 3) but yielded little insight into interactions with the export substrates. Similarly, detecting the interaction of purified SctQ with the effector YopO in vitro required crosslinking, which cannot mimic natural conditions (Extended Data Fig. 2). These results were in line with an inherently transient nature of interactions between secretion systems and their cargo during export. Therefore, we analysed the binding of export substrates to the sorting platform in situ in live *Y. enterocolitica*. Proximity labelling using a translational miniTurbo-SctQ fusion revealed close spatial proximity, indicating direct or indirect interaction of SctQ, not only with the sorting platform component SctL, but also with the majority of the virulence effectors (Table 1 and Supplementary Table 1). Single-particle tracking of fluorescently labelled sorting platform components provided us with detailed, highly internally consistent information about the mobility of complexes formed in the absence or presence of effectors and changes in secreting bacteria (Figs. 1b,c, 2b,d and 3). Presence of effectors and their chaperones reduces the mobility of SctQ and SctL in live bacteria. This effect occurs both in *Y. enterocolitica* with functional injectisomes (Fig. 2b,d) and in bacteria in which no T3SS components other than SctQ/SctL and an effector–chaperone pair are present (Fig. 1). This strongly suggests that effectors directly bind to the central sorting platform proteins SctQ and SctL in the cytosol. For SctQ, the observed changes in the diffusion values are compatible with the stoichiometric binding of one effector to PAmCherry-SctQ (Tables 2 and 3). While our results did not allow us to distinguish between the binding of the effector alone or in complex with a chaperone dimer[62], various studies showing the binding of chaperones to the sorting platform[23,25,27,63,64] argue for the latter possibility. Taken together, these data support a model of direct binding of effectors to SctQ and SctL, which themselves are part of soluble sorting platform complexes. Our data, including the 2:1 stoichiometry of injectisome-bound SctQ:SctL (Extended Data Fig. 3), support an SctK-($Q(Q_C)_2$)$_4$-L$_2$-N complex (see Supplementary Text 5 for extended discussion).

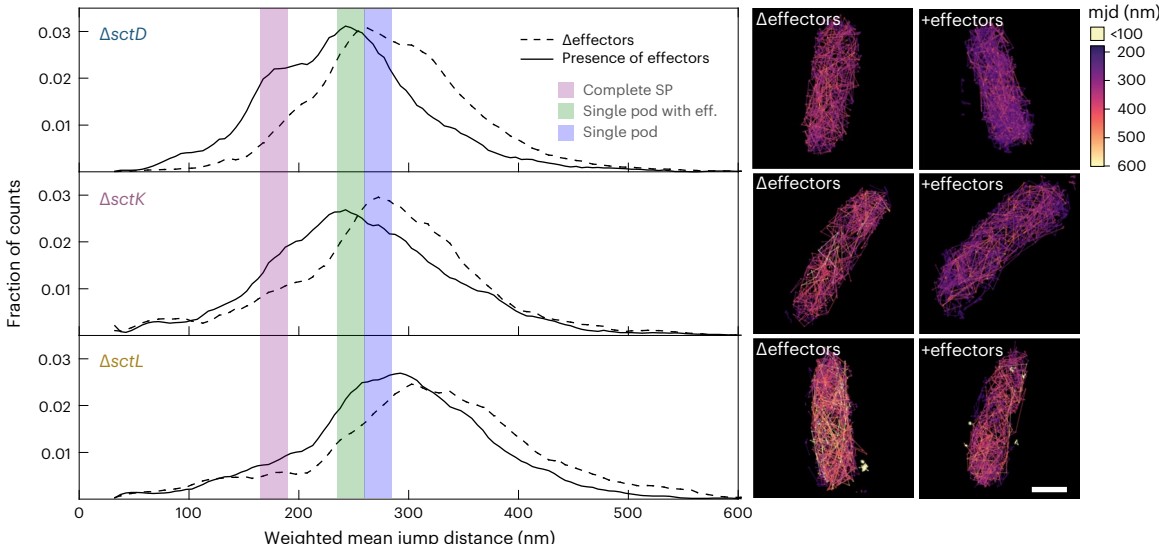

**Fig. 3 | Effector binding to SctQ takes place in the cytosol; while SctD, SctK and SctL are not required for this, SctL enhances binding and pod formation.** Left: mobility of PAmCherry-SctQ in *Y. enterocolitica* lacking the single indicated components of the T3SS in the absence (dashed lines) and presence (solid lines) of the main T3SS effectors. Histogram of mjd of molecular diffusion, weighted for the number of jump distances per trajectory. Vertical bars indicate ranges

for 'single pod' (260–285 nm, blue), 'single pod with effectors' (235–260 nm, green) and 'complete sorting platform' (170–195 nm, purple), as defined in the wild-type strain (Fig. 2, see main text for details). Right and centre: trajectories in representative bacteria with (right) and without (centre) effectors. Scale bar, 0.5 μm. Numbers of trajectories and replications for single-particle tracking experiments are summarized in Supplementary Table 2.

Applying in vivo interaction methods proved essential for determining the transient interactions underlying protein secretion, which are challenging to elucidate by classical methods. Proximity labelling is a suitable method to screen for interaction partners, and sptPALM is a powerful tool to localize and quantify these interactions. Both methods allow for the study of the relevant interactions in situ and do not require protein purification. In addition, analysing live bacteria with the high spatial resolution of PALM yielded additional information on the number and composition of injectisome-bound and soluble sorting platform complexes (Supplementary Texts 3 and 5). Our work clearly highlights that single-particle tracking can be used as a powerful 'in vivo biochemistry' method.

Our demonstration that effectors bind to the sorting platform in live bacteria, together with the exchange of sorting platform proteins at the injectisome[35], supports a shuttling model. In this proposed model, the dynamic cytosolic T3SS components SctK,Q,L,N act as a mobile 'docking complex' for effector binding. This docking complex then delivers the effectors to the core injectisome structure, where they are transferred to another substructure, possibly the export apparatus (Fig. 4). Notably, structural and in situ fluorescence microscopy studies show a high overall occupancy of pod binding sites at the injectisome[7–9,19,28,60,65], which indicates that the exchange rate of sorting platform components is controlled by the dissociation rate and that free binding sites are quickly re-occupied. The biological benefit of this two-step binding process is yet unclear. One plausible benefit is an increased export specificity by the addition of another binding step in a different cellular context. It is known that export by the T3SS is highly specific despite the lack of clear sequence motifs in the unstructured N-terminal secretion signal[66]. The screening of export substrates by the sorting platform, initially using the transient interactions shown in our experiments, may be more efficient in the cytosol than in the highly sterically restricted injectisome-bound form of the sorting platform. Given that the N-terminal T3SS targeting sequence, which is necessary and sufficient for export through the injectisome, enters the secretion channel first, it is possible that binding of the loaded pods to the injectisome triggers the release of the effectors into the space surrounded by the pods, the export apparatus and the ATPase. Effectors can then insert into the export apparatus[58,59]. The shuttling

mechanism might be a way to restrict access of other proteins to this privileged space while efficiently delivering bona fide T3SS substrates, contributing to the specificity of type III secretion. Similar two-step binding processes are known for sec-based protein export[67] and the export of the T4SS relaxase[68,69]. Another non-exclusive potential benefit is an increased binding capacity for high-affinity export substrates before the initiation of secretion. These sorting platform-bound early cargo proteins would then be exported in the first wave of secretion, whereas the ongoing secretion of late cargo proteins afterwards may be largely driven by protein synthesis rates. Overall, this mechanism would contribute to an ordered secretion of effectors. Notably, these interpretations are highly speculative at this moment and will need to be tested in future experiments.

The effector shuttle model can be tested by combining the quantitative data presented in this manuscript with existing quantitative data on the effector secretion rate and the exchange rate of SctQ at the injectisome. For the model to be plausible, the exchange rate of sorting platform subunits at the injectisome must match the rate of effector export. Our results indicate a stoichiometric binding of one effector per SctQ (Tables 2 and 3). This provides a key variable required for examining the effector shuttling model: Based on the previously determined exchange rate of SctQ under secreting conditions (half-time of fluorescence recovery after photobleaching $t_{1/2} = 68.2$ s)[35] and a stoichiometry of 24 SctQ per injectisome[28,29,35] (Extended Data Fig. 3), the SctQ exchange rate per injectisome is ~0.51 s$^{-1}$ (Supplementary Text 6). Assuming one exported effector per exchanged SctQ, this value is compatible with measured export rates (for example, 7–60 effectors s$^{-1}$ per *Salmonella* with 10–100 SPI-1 injectisomes[29,70,71]). Notably, SctQ exchange was measured in the absence of most effectors and under steady-state secreting conditions. In contrast, most export measurements refer to the time directly after activation and to whole bacteria or host cells, rather than single injectisomes[70,72–74]. Also, these calculations did not consider effector binding to SctL (and possibly to SctQ$_C$), which are likely to exchange with injectisome-bound counterparts in subcomplexes with SctQ. Future experiments may include the manipulation of sorting platform exchange at the injectisome and monitoring of the resulting effect on the efficiency of effector export. These studies may reveal whether the exchange of sorting

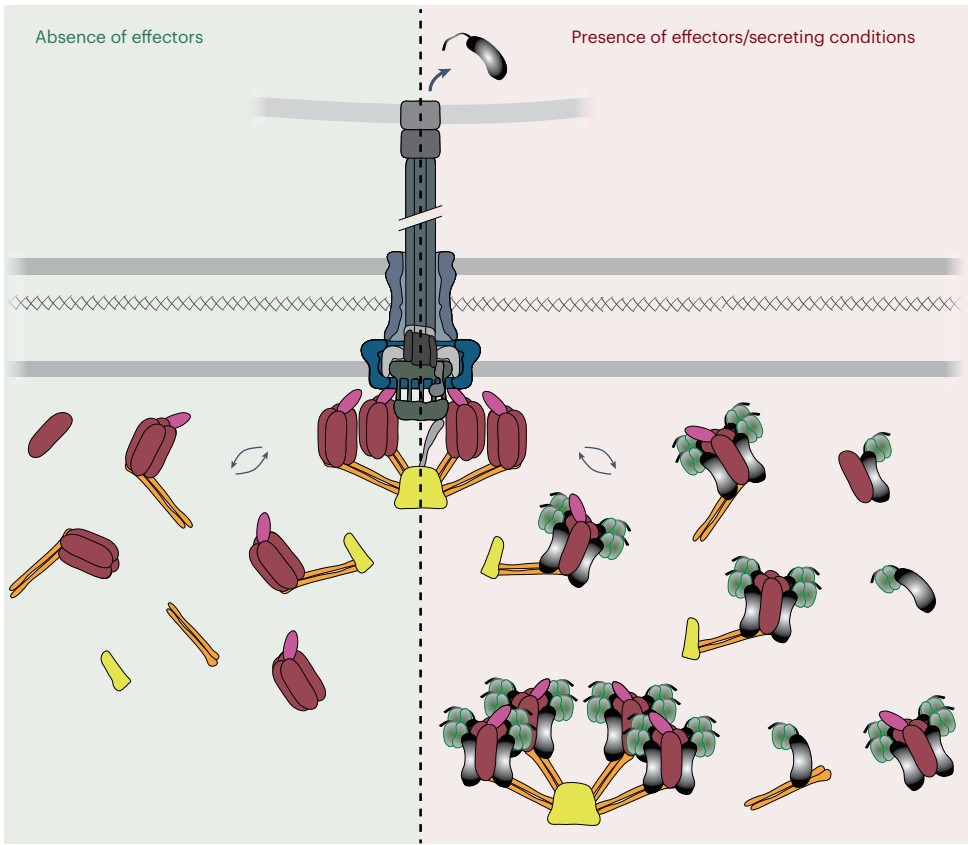

**Fig. 4 | Model of cytosolic sorting platform complexes incorporating the data presented in this study.** Soluble sorting platform complexes in the absence (left) and presence of effectors under secreting conditions (right).

Not all possible complexes are depicted, including complexes not interacting with effector/chaperone complexes on the right side. See main text and Table 2 for details.

platform proteins and their binding to export substrates in the cytosol, as demonstrated in this manuscript, are indeed the basis for an effector shuttle mechanism.

Taken together, our data provide new key insight into the poorly defined initial steps of type III secretion. They reveal that in the bacterial cytosol, T3SS effectors already bind to soluble sorting platform complexes, specifically the core sorting platform components SctQ and SctL, which shuttle between the cytosol and the membrane-spanning injectisome. This mechanism may increase specificity and efficiency of T3SS export and impose an order of export on the different classes of substrates—two critical aspects for the molecular and physiological function of the T3SS. This detailed view into the crucial earliest steps of type III secretion is an essential piece of the puzzle for understanding protein export, and for targeted inhibition and biotechnological application of the T3SS.

## Methods

### Bacterial strain generation and genetic constructs

This manuscript uses the common Sct nomenclature of the T3SS components[75–77]. A list of strains and plasmids used in this study can be found in Supplementary Table 4, and oligonucleotides used are listed in Supplementary Table 5. All *Y. enterocolitica* strains used in this study are based on the *Y. enterocolitica* wild-type strain MRS40 or the strain IML421asd (ΔHOPEMTasd). In IML421asd, all major virulence effector proteins (YopH,O,P,E,M,T) are absent and the strain is referred to as Δeffector in this study. Furthermore, this strain harbours a deletion of the aspartate-beta-semi aldehyde dehydrogenase gene, which renders the strain auxotrophic for diaminopimelic acid (DAP), making it suitable for work in a biosafety class 1 environment[48]. To

ectopically express proteins in *Y. enterocolitica*, the corresponding genes were cloned into expression plasmids. The plasmid sequences were confirmed by sequencing, and plasmids were transformed into the respective bacteria by electroporation. To co-express the effector YopO-Flag and its chaperone SycO from plasmid, SycO and YopO were amplified from the pYV virulence plasmid, adding a C-terminal Flag tag sequence to YopO to allow detection with an anti-Flag antibody. All other fusion proteins used in situ in this study are expressed as endogenous translational fusions introduced into the native genetic background by allelic exchange[43].

### Bacterial cultivation, secretion assays and protein analysis

*Y. enterocolitica* BHI (brain heart infusion broth) day cultures supplemented with nalidixic acid (35 mg ml⁻¹), glycerol (0.4%), MgCl₂ (20 mM) and DAP (60 μg ml⁻¹), where required, were inoculated from stationary overnight cultures to an optical density (OD)$_{600}$ of 0.15 for secreting and 0.12 for non-secreting conditions. Where required, ampicillin (200 μg ml⁻¹) or chloramphenicol (10 μg ml⁻¹) was added to select for the maintenance of the respective expression plasmids. For secreting conditions, 5 mM EGTA was added; for non-secreting conditions, 5 mM CaCl₂ was added to the pre-warmed (~55 °C) medium, which was filtered through 0.22 or 0.45 μm filters before the addition of other supplements. Unless indicated otherwise, after inoculation, day cultures were incubated at 28 °C for 90 min to allow the bacteria to reach exponential growth phase. Expression of the *yop* regulon including the T3SS machinery genes was then induced by a rapid temperature shift to 37 °C in a water bath. Where indicated, protein expression from plasmid was induced at this point by the addition of L-arabinose (0.2%, unless indicated differently).

For protein secretion assays and analysis of total cellular proteins, bacteria were incubated for 3 h at 37 °C. Of the culture, 2 ml was collected after centrifugation at 21,000 g for 10 min. The supernatant was removed from the total cell pellet and proteins were precipitated by addition of a final concentration of 10% trichloroacetic acid and incubation at 4 °C for 1–8 h. Precipitated proteins were collected by centrifugation for 15 min at 21,000 g and 4 °C. The pellet was washed once with 1 ml ice-cold acetone and subsequently resuspended and normalized in SDS–PAGE loading buffer. Total cellular protein samples were normalized to 0.3 OD units (ODu; 1 ODu is equivalent to 1 ml of culture at on OD$_{600}$ of 1, corresponding to ~5 × 10$^8$ Y. enterocolitica cells) per 15 µl, which is the volume loaded into SDS–PAGE gels. Supernatant samples were normalized to 0.6 ODu per 15 µl. Samples were incubated for 10 min at 95 °C. Separation was performed on 11–15% SDS–PAGE gels using Precision Plus All Blue Prestained (Biorad) or Blue Classic Prestained Marker (Jena Biosciences) as a size standard. For visualization, the SDS–PAGE gels were stained with Instant Blue (Expedeon). For immunoblots, the separated proteins were transferred from the SDS–PAGE gel onto a nitrocellulose membrane. Primary rabbit antibodies against mCherry (Biovision 5993, 1:2,000) or against the Flag peptide (Rockland, 600-401-383S, 1:5,000) were used in combination with secondary anti-rabbit antibodies conjugated to horseradish peroxidase (HRP) (Sigma, A8275, 1:5,000); for the detection of biotinylated proteins, a streptavidin-HRP conjugate was used (Amersham, RPN1231, 1:40,000). For visualization, ECL chemiluminescence substrate (Millipore, WBLUF0500) was used in a LAS-4000 Luminescence image analyser.

## Co-immunoprecipitation using shotgun proteomics

For co-immunoprecipitation (Extended Data Fig. 1), 100 ml of non-secreting culture medium were inoculated to an OD$_{600}$ of ~0.15 and treated as described above. After 2.5 h incubation at 37 °C, the cultures were transferred to 50 ml tubes and centrifuged at 2,000 g for 15 min at 4 °C. Pellets were resuspended in 10 ml of cold phosphate-buffered saline (8 g l$^{-1}$ NaCl, 0.2 g l$^{-1}$ KCl, 1.78 g l$^{-1}$ Na$_2$HPO$_4$·2H$_2$O, 0.24 g l$^{-1}$ KH$_2$PO$_4$, pH 7.4); the contents of two tubes were pooled and centrifuged at 2,000 g for 15 min at 4 °C. The pellet was resuspended in 10 ml HNN lysis buffer (50 mM HEPES pH 7.5, 150 mM NaCl, 50 mM NaF, 1 tablet per 50 ml of cOmplete Mini EDTA-free protease inhibitor (Roche, 11836170001), filter-sterilized) and frozen at −20 °C overnight. On the next day, samples were thawed on ice and cells were mechanically lysed using a French Press (G. Heinemann) in a THU-600 BIG 20k cell with an opening pressure of 1,241.1 bar. The procedure was repeated 3–4 times until cell lysates were clear. Afterwards, the bacterial lysate was centrifuged at 4,600 g at 4 °C for 45 min to remove insoluble debris.

For affinity purification, 10 µl of bead slurry (RFP-Trap Magnetic Agarose, Chromotek) was added to the lysate, which was then incubated for 1 h at 4 °C on a turning wheel. Beads were captured from the supernatant with a magnetic rack, washed twice with 1 ml HNN lysis buffer and reconstituted in 100 µl HNN lysis buffer. Elution was performed by heat separation for 10 min at 95 °C in 2x SDS–PAGE loading buffer (4% sodium dodecyl sulfate (SDS), 0.2 M Tris, 20% glycerol, 0.1 M dithiothreitol, bromophenol blue, pH 6.8). To monitor the experimental steps, samples were taken throughout the whole procedure and analysed by immunoblotting.

To remove the SDS, the eluate was precipitated using 7 volumes of acetone for 2 h at −20 °C. Following centrifugation, the protein pellet was washed twice with 300 µl ice-cold methanol, dried and then reconstituted in 100 mM ammonium bicarbonate containing 1 mM Tris(2-carboxyethyl)phosphine (TCEP) with additional incubation at 90 °C for 10 min to reduce disulfide bonds. Alkylation of TCEP-reduced disulfide bonds was performed with 5 mM iodoacetamide at 25 °C for 30 min in the dark. Trypsin (1 µg) was added for protein digestion carried out at 30 °C overnight. The samples were then acidified using trifluoroacetic acid and peptides were purified using Chromabond C18 microspin columns (Macherey-Nagel).

Peptide mixtures were analysed using LC–MS carried out on an Exploris 480 instrument connected to an Ultimate 3000 RSLC nano with a Prowflow upgrade and a nanospray flex ion source (all Thermo Scientific). Peptide separation was performed on a reverse phase HPLC column (75 µm × 42 cm) packed in-house with C18 resin (2.4 µm, Dr Maisch). The following separation gradient was used: 94% solvent A (0.15% formic acid) and 6% solvent B (99.85% acetonitrile, 0.15% formic acid) to 35% solvent B over 40 min. The data were acquired in data dependent acquisition mode with the following settings: 1 MS scan at a resolution of 60,000 with 25 ms maximum ion injection fill time and 300% AGC target settings, MS/MS resolution at 15,000 scans with 50 ms max. fill time and a cycle time of 1 s. AGC target settings were 200% and HCD collision energy set to 27%.

For all label-free quantification, the MS raw data were then analysed with MaxQuant at standard settings using a protein database containing proteins of the closely related Y. enterocolitica strain W22703 and of the pYVe227 virulence plasmid (GenBank entry AF102990.1). Statistical analysis of the MaxQuant LFQ data was performed on an updated SafeQuant R-script to routinely process MaxQuant 'protein groups' outputs[78].

## Proximity labelling using miniTurbo and shotgun proteomics

For proximity labelling experiments (Table 1 and Supplementary Table 1), bacteria were incubated as described above with the following changes: Culture volumes were increased to 20 ml, and 100 µM biotin and 0.1% arabinose (for expression of the EGFP-miniturbo control at a level comparable to the native miniTurbo-SctQ levels; Supplementary Fig. 11) were added at the time of the temperature shift to 37 °C. Cell pellets from 15 ml culture were normalized by culture to OD$_{600}$, collected by centrifugation (4,686 g, 4 °C, 8 min) and stored at −20 °C. Pellets were thawed on ice, resuspended in 200 µl lysozyme buffer (50 mM Tris pH 7.8, 12 mM EDTA, 1.2 mg ml$^{-1}$ lysozyme) and incubated at 4 °C. After 60 min, 200 µl of detergent cocktail (8% 3-[(3-cholamidopropyl) dimethylammonio]-1-propanesulfonate hydrate (CHAPS), 8% 3-[dimethyl(tetradecyl)azaniumyl] propane-1-sulfonate (zwittergent 3-14), 8% sodium lauroyl sarcosinate (SLS), 0.1% Nonidet P-40 (NP-40), 40% glycerol) and 40 µl protease inhibitor mix (1 tablet cOmplete Mini EDTA-free protease inhibitor (Roche, 11836170001) in 1 ml 50 mM Tris-HCl pH 7.8) were added. The resulting bacterial suspensions were lysed by sonication (Hielscher, UP200St) until the samples were clear. Unlysed cells and other insoluble parts were removed by centrifugation at 20,000 g for 20 min at 4 °C. The cleared lysate was transferred to purification buffer (50 mM Tris-HCl pH 7.8, 150 mM NaCl, 0.1% Triton X-100) using PD MiniTrap Sephadex G-25 columns (Cytiva, 28-9180-07) and the volume was adjusted to 10 ml. Biotinylated proteins were separated by binding to 40 µl magnetic streptavidin beads (Pierce, 88817) on a rotor for 1 h. The beads were washed three times in purification buffer and three times in purification buffer without detergent. Beads were then resuspended in 100 µl 2% SLS and incubated at 99 °C for 12 min. This procedure was repeated and the two eluate fractions were pooled. The pooled eluate was then precipitated using 7 eluate volumes of acetone for at least 2 h at −20 °C. Following centrifugation, the protein pellet was washed twice with 300 µl ice-cold acetone. The protein pellet was dried, reconstituted in 100 mM ammonium bicarbonate containing 1 mM TCEP and incubated at 90 °C for 10 min. Alkylation of reduced disulfide bonds was performed with 5 mM iodoacetamide at 30 min at 25 °C in the dark. Trypsin (1 µg) was added for protein digestion carried out at 30 °C overnight. Further sample preparation and analytical steps were carried out as described above with identical instrumentation and software settings.

## Total proteome analysis using shotgun proteomics

For total proteome analysis, cell pellets were resuspended in 2% SLS in 100 mM ammonium bicarbonate and heated for 15 min at 90 °C. Proteins were then reduced by adding 5 mM TCEP at 95 °C for 15 min,

followed by alkylation (10 mM iodoacetamide, 30 min at 25 °C). Total protein (50 µg) was then digested with 1 µg trypsin (Serva) overnight at 30 °C. Following digestion, SLS was precipitated with trifluoroacetic acid (1.5% final concentration) and peptides were purified using Chromabond C18 microspin columns (Macherey-Nagel).

For the data presented in Supplementary Table 3, purified peptides were analysed using LC–MS as previously reported[79]. Briefly, peptides were separated as described above, but with 90 min separation at a flow rate of 300 nl min[−1]. MS data were acquired in a Q Exactive Plus mass spectrometer (Thermo Fisher Scientific) with the following settings: 1 MS scan at a resolution of 70,000 with 50 ms max. ion injection fill time, MS/MS at 17,500 scans of the 10 most intense ions with 50 ms maximum fill time. The data were further analysed using MaxQuant[78] and SafeQuant[80].

Due to an instrumental upgrade, total proteome samples presented in Supplementary Fig. 5 were analysed on an Exploris 480 connected to an Ultimate 3000 RSLC nano. The LC peptide separating gradient was reduced to 60 min (6–35% buffer B). The MS data were acquired in data independent acquisition mode (DIA) using 45 windows with an isolation window of 14 $m/z$ with 1 $m/z$ overlap (see ref. 81 for details). MS scan resolution was set to 120,000 (MS1) and 15,000 (DIA) with a scan range of 350–1,400 $m/z$ (MS1) and 320–950 precursor mass range (DIA). AGC target settings were 300% (MS1) and 3,000% (DIA) with a maximum ion injection time of 50 ms (MS1) and 22 ms (DIA).

DIA data were analysed using DIA-NN (v.1.8)[82] and a *Y. enterocolitica* protein database containing proteins of the closely related *Y. enterocolitica* strain W22703 and the pYVe227 virulence plasmid (GenBank entry AF102990.1). Full tryptic digest was allowed with two missed cleavage sites, and oxidized methionine residues and carbamidomethylated cysteine residues. 'Match between runs' and 'remove likely interferences' options were enabled. The neural network classifier was set to the single-pass mode, and protein inference was based on genes. Quantification strategy was set to any LC (high accuracy). Cross-run normalization was set to RT-dependent. Library generation was set to smart profiling. DIA-NN outputs were further evaluated using R.

### Protein purification for in vitro interaction studies

Proteins were expressed in *Escherichia coli* BL21(DE3) harbouring pCDFduet- and pET24b-based expression plasmids (Supplementary Table 4). Fresh 200 ml LB liquid media containing 50 µg ml[−1] chloramphenicol and 5 µg ml[−1] kanamycin (for pET24b-based plasmids) or 50 µg ml[−1] streptomycin (for pCDFduet-based plasmids) were inoculated with 2 ml of stationary overnight cultures. The cultures were incubated at 37 °C with shaking (180 r.p.m.) until they reached an $OD_{600}$ of 0.4–0.6 (usually after 3–5 h). At this point, the expression of the proteins was induced with 0.5 mM isopropyl β-D-1-thiogalactopyranoside (IPTG), and the cultures were incubated for 12–14 h at 18 °C with shaking (180 r.p.m.). After incubation on ice for 15 min, bacteria were collected by centrifugation (4,000 $g$, 15 min, 4 °C) and the pellet was washed twice with 25 ml ice-cold filtered PBS. After the last wash, the pellet was resuspended in 10 ml of ice-cold lysis buffer (50 mM $NaH_2PO_4$ pH 7.4, 150 mM NaCl, 1 mM EDTA, 1 mg ml[−1] lysozyme, 5% glycerol, 1% Triton X-100, 1 tablet per 50 ml of cOmplete Mini EDTA-free protease inhibitor (Roche, 11836170001)) and incubated on a rotatory shaker for 1 h at 4 °C. The cells were mechanically lysed by passing the suspension through a French Press as described above for up to five times until the samples become clear. The crude lysate was then centrifuged at 4,600 $g$ for 1 h or at 10,000 $g$ for 30 min at 4 °C to remove insoluble debris. The soluble fraction was filtered through a syringe using a 0.22 µm filter to remove any unlysed bacterial cells.

His-tagged fusion proteins were affinity purified by incubating the supernatant with 0.5 g of Ni-IDA resin (Protino, Macherey-Nagel, 745210.120) on a rotary shaker for 1 h at 4 °C. The supernatant was removed by centrifugation at 500 $g$ for 1 min. The resin was washed three times with 5 bed volumes of ice-cold LEW buffer (lysis-elution-wash: 50 mM $NaH_2PO_4$ pH 7.4, 150 mM NaCl, 1 mM lysozyme, 5 mM beta-mercaptoethanol, 1 mM dithiothreitol (DTT),

5% glycerol) supplemented with 10 mM imidazole. Elution buffer (2 ml, LEW with 250 mM imidazole) was then added to the resin and the resin incubated for 5 min at 4 °C. The eluate was collected after centrifugation at 500 $g$ for 1 min, subsequently transferred to LEW buffer using PD MiniTrap Sephadex G-25 columns (Cytiva, 28-9180-07) and concentrated using Amicon Ultra 0.5 ml centrifugal filter units (Merck). Flag-tagged fusion proteins were purified by incubating the soluble cell lysate with pre-equilibrated anti-Flag M2 affinity gel (100 µl−1 ml) for 2 h at 4 °C on a rotatory shaker. After incubation, the samples were centrifuged at 1,000 $g$ for 5 min at 4 °C to collect the resin. The purification was followed by six washes using 5 ml wash buffer (50 mM $NaH_2PO_4$ pH 7.4, 600 mM NaCl, 5% glycerol, 1% Triton X-100) and centrifugation at 1,000 $g$ for 5 min at 4 °C. After the last wash, the Flag-tagged proteins were eluted in elution buffer (50 mM $NaH_2PO_4$ pH 7.4, 150 mM NaCl, 150 ng ml[−1] of 3x Flag peptide). Two resin volumes of elution buffer were added to the resin and the resuspension was incubated for 30 min on a rotating shaker at 4 °C. The eluate was collected by centrifugation at 4,600 $g$ for 30 s at 4 °C and transferred to a fresh microcentrifuge tube. Glycerol was added to the eluate to obtain a final concentration of 5% and the samples were rapidly frozen using liquid nitrogen. Before interaction experiments, the eluate fractions were concentrated in storage buffer (20 mM $NaH_2PO_4$ pH 7.4, 200 mM NaCl, 5% glycerol and 1 mM DTT) using Amicon Ultra 0.5 ml centrifugal filter units (Merck). Both His-tagged and Flag-tagged purified proteins were flash-frozen and stored at −70 °C until further use.

### Interaction measurements using biolayer interferometry

In vitro equilibrium interaction measurements were performed on a BLItz instrument (ForteBio) at room temperature. Ni-NTA biosensors were pre-hydrated for 10 min using BLI buffer (50 mM $NaH_2PO_4$, 150 mM NaCl, 5 mM beta-mercaptoethanol, 1 mM DTT, 5% glycerol, pH 7.4). Increasing concentrations of purified His-YopO and SycO (1, 5, 20 µM) were separately immobilized on the Ni-NTA biosensor for 120 s. After the loading step, the biosensor was dipped into a tube containing 250 µl of BLI buffer for 30 s to remove excess analyte. Next, the biosensor was exposed to the target protein (4 µl of SctQ-Flag; 5, 10, 33 µM) for 120 s (association step), followed by a wash with BLI buffer for 120 s (dissociation step). EGFP-His was used as the negative control in this experiment. Data were recorded and sensorgrams were generated using BLItz Pro software v.1.1.0.31 (Pall ForteBio).

### In vitro crosslinking interaction experiments

For in vitro crosslinking, the purified proteins were diluted to a final concentration of 5 µM in 200 µl of equilibrium buffer (20 mM $NaH_2PO_4$, 200 mM NaCl, 5% glycerol, 1 mM DTT, pH 7.4). Of the His-tagged protein, 200 µl (5 µM) were combined with 200 µl of the respective Flag-tagged protein (5 µM) and 1% formaldehyde was added to stabilize transient interactions. The mixture was incubated on a rotatory shaker at 4 °C for the indicated time periods. At the end of the incubation, the crosslinking reaction was quenched using 180 µl of 1 M Tris-HCl pH 8.0. To purify the His-tagged protein after crosslinking, 40 µl of Ni-IDA suspension (50 mg ml[−1] of Ni-IDA suspended in 50 mM $NaH_2PO_4$ pH 7.4, 500 mM NaCl, 5% glycerol, 1 mM DTT, 5 mM β-mercaptoethanol and 20 mM imidazole) were added to each sample and the sample incubated for 30 min at 4 °C on a rotatory shaker. The resin was collected by centrifugation (500 $g$, 1 min, 4 °C) and washed six times with 400 µl of equilibrium buffer. Elution buffer (200 µl; 50 mM $NaH_2PO_4$ pH 7.4, 500 mM NaCl, 5% glycerol, 1 mM DTT, 5 mM β-mercaptoethanol and 250 mM imidazole) was added to the washed Ni-IDA resin and the resin incubated for 10 min at 4 °C on a rotatory shaker. The eluate was removed by centrifugation (500 $g$, 1 min, 4 °C).

### Wide-field fluorescence microscopy

For fluorescence microscopy, bacteria were treated as described above. After 2–3 h at 37 °C under secreting conditions, 400 µl of bacterial

culture were collected by centrifugation (2,400 $g$, 2 min) and reconstituted in 200 µl minimal microscopy medium (100 mM HEPES pH 7.2, 5 mM $(NH_4)_2SO_4$, 100 mM NaCl, 20 mM sodium glutamate, 10 mM $MgCl_2$, 5 mM $K_2SO_4$, 50 mM 2-($N$-morpholino)ethanesulfonic acid, 50 mM glycine, 60 µg ml$^{-1}$ DAP, 5 mM EGTA). Of the resuspension, 2 µl were spotted on agarose pads (1.5% low-melting agarose (Sigma-Aldrich) in minimal microscopy medium) in glass depression slides (Marienfeld). Microscopy was performed on a Deltavision Elite optical sectioning microscope equipped with an UPlanSApo ×100/1.40 oil objective (Olympus), using an Evolve EMCCD camera (Photometrics). Exposure times were 200 ms for GFP fluorescence, using a GFP filter set (475/28 nm excitation and 525/48 nm emission filter sets) and 25 ms for bright field. Per image, a $z$ stack containing 11 frames per wavelength with a spacing of 150 nm was acquired. The micrographs where subsequently deconvolved using softWoRx 7.0.0 (standard 'conservative' settings). Images were further processed with FIJI (ImageJ 1.51f/1.52i/1.52n)[83]. For presentation of micrographs in a figure, representative fields of view at central $z$ levels were selected.

## Protein stability assay

For the stability assay presented in Supplementary Fig. 5, bacteria were grown under non-secreting conditions as described above. After 3 h at 37 °C, the $OD_{600}$ was measured and normalized to 2.5. Tetracycline (10 µg ml$^{-1}$) was added to the cultures to stop the protein biosynthesis, and cultures were continuously incubated at 37 °C. Samples (1 ml) were taken directly before the addition of tetracycline and at 10, 30 and 60 min afterwards to monitor protein stability. Samples were analysed by total proteome analysis, as described above, and protein stability was determined as the ratio of intensity values before and after addition of tetracycline.

## sptPALM imaging

*Y. enterocolitica* cells were cultured as described above. After incubation at 37 °C for 2.5 h in BHI medium, 500 µl of bacterial culture were centrifuged for 3 min at 4,000 $g$ at 37 °C and resuspended in 200 µl pre-warmed (37 °C) minimal microscopy medium containing the same supplements. Bacteria were then incubated at 37 °C with shaking for 30 min and washed four times with 500 µl pre-warmed (37 °C) EZ medium (Teknova) supplemented with DAP and either 5 mM $CaCl_2$ or 5 mM EGTA for non-secreting conditions or secreting conditions, respectively. After the final washing step, cells were concentrated by resuspending them in 100–200 µl of EZ medium with the supplements described above and spotted on precast pads of 1.5% low-melting agarose in EZ medium, again with the supplements described above, on a KOH-cleaned microscopy slide in an enclosed area (Thermo Fisher Scientific GeneFrame, AB-0577). For fixed-cell experiments, cells were fixed with 2% paraformaldehyde (Sigma-Aldrich) for 15 min and washed three times with EZ medium before spotting them on the microscopy slide. SptPALM experiments were performed on a custom-built setup based on an automated Nikon Ti Eclipse microscope housing appropriate dichroic and filters (ET DAPI/FITC/Cy3 dichroic, ZT405/488/561rpc rejection filter, ET610/75 bandpass, Chroma) and a CFI Apo TIRF ×100 oil objective (NA 1.49, Nikon). All lasers (405 nm OBIS, 561 nm OBIS; both from Coherent) were modulated via an acousto-optical tunable filter (AOTF, Gooch and Housego). Fluorescence signal was detected by an EMCCD camera (iXON Ultra 888, Andor) at a pixel size of 129 nm, in frame transfer mode and readout parameter settings of EM-gain 300, pre-amp gain 2 and 30 MHz readout speed. The $z$-focus was controlled by a commercial perfect focus system (Nikon). Image acquisitions were controlled using Micro-Manager (v.1.4.23)[84]. All live cell sptPALM experiments were performed on a custom-built heating stage at 25 °C. Live *Y. enterocolitica* cells were imaged in HILO illumination mode[85]. Applied laser intensities measured after the objective were 35 W cm$^{-2}$ (405 nm) and 800 W cm$^{-2}$ (561 nm). Before the acquisition of each sptPALM video, a pre-bleaching step of 561 nm illumination (800 W cm$^{-2}$)

was applied for 30 s to the corresponding region of interest to reduce autofluorescence. sptPALM videos were then recorded for 20,000 frames, pulsing the 405 nm laser every 10th imaging frame at 67 Hz with an exposure time of 15 ms per frame, while the 561 nm laser was continuously active. After each image acquisition, a bright light snapshot of the corresponding region was recorded to obtain bacterial cell shapes. For stoichiometry experiments of SctQ and SctL in live and fixed cells, sptPALM movies were recorded for 80,000 frames to ensure a full readout of PAmCherry molecules.

## sptPALM post-processing and data analysis

Bacterial cell shapes were extracted from bright light snapshots by manually segmenting single bacterial cells using Fiji (ImageJ 1.51 v/1.52p/1.53c)[83]. Raw sptPALM movies were processed using rapidSTORM (3.3.1)[86] to obtain single-molecule localizations, fluorescence intensities and timepoints of each localization from each individual movie, which were saved in localization files. These localization files from individual sptPALM movies were then split into localization files for each individual bacterial cell using a custom-written Python script and the previously extracted bacterial cell shapes. To reduce processing load, localizations for each individual cell were tracked, visualized and filtered using the custom-written 'swift' tracking software 0.3.1. Obtained tracked single-cell localization files originating from the same sptPALM movie were then merged, creating a tracked localization file for each individual sptPALM movie. For tracking of single-molecule localizations, trajectories were allowed to have a maximum of 3 frames of dark time (for example, caused by fluorophore blinking) and the mjd was calculated for trajectory segments with more than 6 one-frame jumps and less than 31 one-frame jumps (jumps spanning several frames due to dark times were not used in mjd calculations). Mean jump distances were weighted according to the number of jump distances per trajectory segment. Where only the cytosolic fraction of trajectories was analysed, membrane-bound trajectories were excluded from the analysis by overlaying the corresponding bright light image. For each trajectory, the mjd was weighted by the number of data points used for the mjd calculation (mjd_n) and plotted in a line diagram using 'Origin' 2019.

## Quantification of fluorescent foci

Obtained trajectory segments were assigned to their diffusive states (static and mobile) in 'swift' on the basis of the experimental localization precision of ~25 nm (determined using the NeNA method[87]) and filtered for trajectories assigned to the static diffusive state. To ensure that only membrane-bound trajectories were counted, the corresponding bright light image was overlaid and only foci overlapping with the cell membrane were included in the analysis. Individual foci were then counted manually and plotted in a box diagram using Origin 2019.

The number of trajectories per focus was then quantified by extracting the 'track.ids' in 'swift' from each previously selected focus, as the 'track.id' is a unique value assigned to each individual trajectory. The numbers of individual 'track.ids' were then plotted in a box diagram using Origin 2019. For all three tested conditions, the determined numbers of trajectories were then summarized in one dataset, displayed in histograms with a bin size of 1 and then fitted using a custom-written script in Python (v.3.8.5) with the binomial distribution probability mass function given as:

$f(k, n, p) = \binom{n}{k} p^k q^{n-k}$, where $n$ is the sample size, $p$ is the success probability, $k$ the number of successes and $q$ the number of failures. The sample size $n$ was defined as the expected number of PAmCherry molecules per injectisome ($n = 24$), $k$ as the number of successfully detected PAmCherry molecules per injectisome and our fitting interval as the range from $k = 1$ to $k = 24$, as by definition we cannot measure $k = 0$. Using the obtained value for $p$, the frequency of missing all 24 PAmCherry molecules per injectisome during detection, given as $f(0, 24, p)$, was calculated using Excel 2019. Previously determined numbers

of injectisomes per cell were then corrected by dividing the average numbers by $1 - f(0, 24, p)$.

### Measurements of SctQ and SctL stoichiometry in live and fixed cells

For quantifying the copy numbers of SctQ and SctL per injectisome, full readout sptPALM experiments were performed and recorded data were tracked as described above. Corresponding bright light images were overlaid to ensure the correct selection of membrane-bound foci. The number of trajectories per focus was then again determined by extracting the 'track.ids' from each selected focus and displayed in histograms in Origin 2019, with a bin size of 1 for the SctL datasets and a bin size of 2 for the SctQ datasets. The resulting histograms were then fitted with the binomial distribution probability mass function as described above. Here, the SctL datasets were fitted for a stoichiometry of $N = 12$, and SctQ datasets were fitted for $N = 6, 12$ or $24$. To evaluate the goodness of fit for each dataset, the squared correlation coefficient ($r^2$) of the histograms and binomial fits were calculated using Excel 2019.

### Statistical comparison of protein mobility distributions

To compare the protein mobility distributions of labelled proteins in different strain backgrounds and under different conditions, we calculated the sum of squared differences ($\Sigma(\Delta^2)$) of the cumulative distributions and the squared correlation coefficient ($r^2$) of the distributions of the mean jump distances determined by sptPALM. $\Sigma(\Delta^2)$ of 0 and $r^2$ of 1 indicate identical distributions, whereas higher $\Sigma(\Delta^2)$ and lower $r^2$ indicate differences in the distributions. Note that $\Sigma(\Delta^2)$ scales with the number of measurements, which was constant within individual experiments.

### Reporting summary

Further information on research design is available in the Nature Portfolio Reporting Summary linked to this article.

## Data availability

All relevant data are included in the paper and/or its supplementary files. The mass spectrometry proteomics data have been deposited to the ProteomeXchange Consortium via the PRIDE partner repository with the dataset identifier PXD044214 (see Supplementary Table 6 for assignment). Supplementary videos are accessible at https://doi.org/10.17617/3.HMABQ2. Source data are provided with this paper.

## Code availability

Rapidstorm software v.3.3.1 was used for single-molecule localization. ImageJ1.51 v./1.52p/1.53c based Fiji software package was used for cell segmentation. Custom-written tracking software *swift* was used for tracking, visualizing and filtering sptPALM data (http://bit.ly/swifttracking; v.0.3.1, used in this manuscript, and all subsequent versions of the software can also be obtained upon request to the authors). Binomial distribution fit analysis was performed with Python v.3.8.10 using the SciPy optimize library. Plotting of data was performed using Origin 2019. Protein data were analysed with MaxQuant[78] in standard settings. If further statistical analysis was required, the MaxQuant 'proteinGroups.txt' file was further evaluated using SafeQuant 2.3.5 (ref. 90, https://github.com/georgiaAngelidou/SafeQuant.v2.3.5, current versions including DIA-NN output processing are available upon request to the authors). Standard spreadsheet calculations were performed using Microsoft Excel 2016.

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

## Acknowledgements

This work was supported by the Max Planck Society, the Deutsche Forschungsgemeinschaft (grant DI 1765/5-1 to A.D.), and the German Academic Exchange Service (DAAD) (PPP USA 2021-2023 to A.D. and U.E.). We thank L. Weber, University of Bonn, for support with the binomial fitting analysis to determine the number of injectisomes per bacterium; G. Angelidou, Max Planck Institute for Terrestrial Microbiology, Marburg, for support with the statistical analysis of the proteomics data; H. Niemann, University of Bielefeld, for critical reading of the manuscript; A. Harms and M. Herfurth, Max Planck Institute for Terrestrial Microbiology, Marburg for support with the in vitro experiments and proximity labelling, respectively; and T. Zhou, Vanderbilt University, for language editing.

## Author contributions

S.W. performed experiments, assisted in data analysis and participated in study design and writing of the manuscript. A.B. operated the PALM microscope, performed the majority of sptPALM data analysis and participated in writing the manuscript. C.B. performed protein purification, interaction experiments and assisted with the co-immunoprecipitation experiments. K.P. performed protein purification and interaction experiments. J.V. performed proximity labelling experiments. B.T. established protocols and contributed code for data analysis. C.H., M.F. and K.L. assisted in experiments. T.G. and J.K. from the Proteomics facility provided this technology and assisted in data evaluation and statistical analysis. U.E. provided supervision, performed data analysis, participated in study design and in writing the manuscript. A.D. conceived and designed the study, provided supervision, performed data analysis and wrote the manuscript.

## Funding

## Competing interests

The authors declare no competing interests.

## Additional information

**Extended data** is available for this paper at https://doi.org/10.1038/s41564-023-01545-1.

**Correspondence and requests for materials** should be addressed to Ulrike Endesfelder or Andreas Diepold.

**a**

| | T3SS machinery | | | | | | | | Chaperones | |
|---|---|---|---|---|---|---|---|---|---|---|
| | *Sorting platform* | | | | *Membrane rings* | | | *E.ap.* | | |
| | SctQ | SctL | SctN | SctK | SctC | SctD | SctJ | SctV | SycD | YscG |
| PAmCh-**SctQ** | +++ | +++ | | | | | ++ | | + | + |
| PAmCh-**SctL** | ++ | +++ | | + | + | | | + | + | ++ |
| PAmCh-**SctN** | | + | +++ | | | + | ++ | + | + | + |

| | Export cargo | | | | | | | | | |
|---|---|---|---|---|---|---|---|---|---|---|
| | *Ruler* | *Translocon* | | | *Effector proteins* | | | | | |
| | SctP | SctA | SctB | SctE | YopO | YopQ | YopP | YopM | YopH | YopE |
| PAmCh-**SctQ** | | + | + | | + | + | + | | | |
| PAmCh-**SctL** | + | | | | ++ | ++ | + | ++ | | |
| PAmCh-**SctN** | ++ | | | | + | + | | | | |

**b**

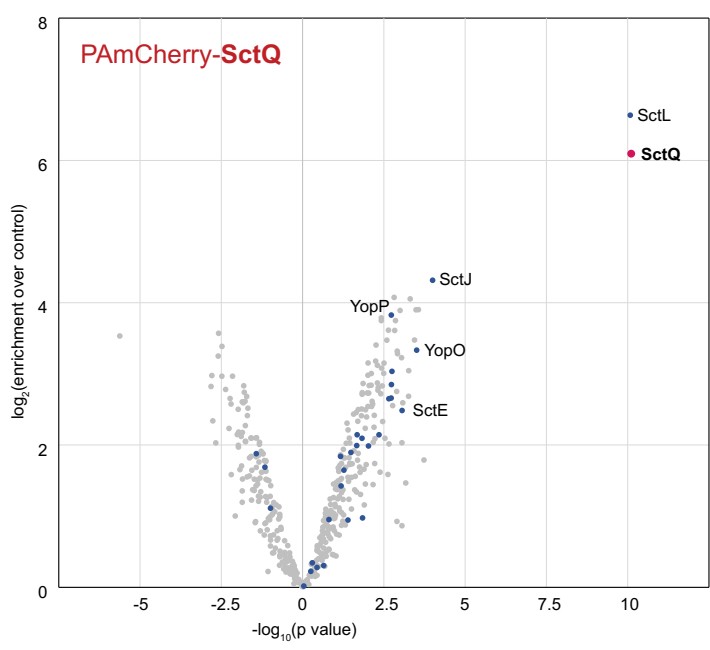

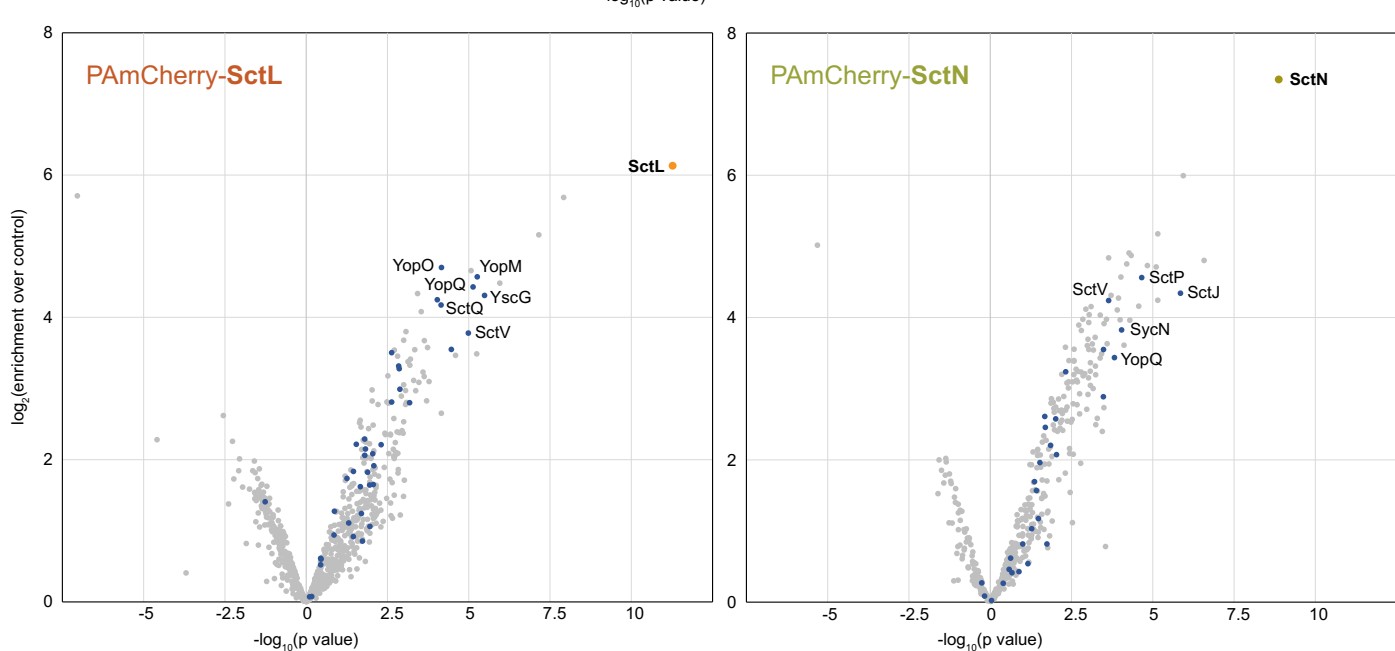

**Extended Data Fig. 1 | See next page for caption.**

**Extended Data Fig. 1 | Co-immunoprecipitation of labeled sorting platform components in live bacteria. a**) Summary of the significance of enrichment of selected T3SS machinery components (e.ap., export apparatus), effectors and chaperones upon co-immunoprecipitation of the sorting platform proteins SctQ, SctL, SctN tagged with PAmCherry (PAmCh). +++/++/+ = strong/ intermediate/weak significance of enrichment (both $\log_2$(enrichment over control) and $-\log_{10}(p) > 6/4/2$ in homoscedastic two-tailed t-tests, respectively; see Material and Methods for details). **b**) Corresponding volcano plots. Bait proteins in bold, T3SS components marked by blue dots, names of enriched T3SS components indicated. $n = 3$.

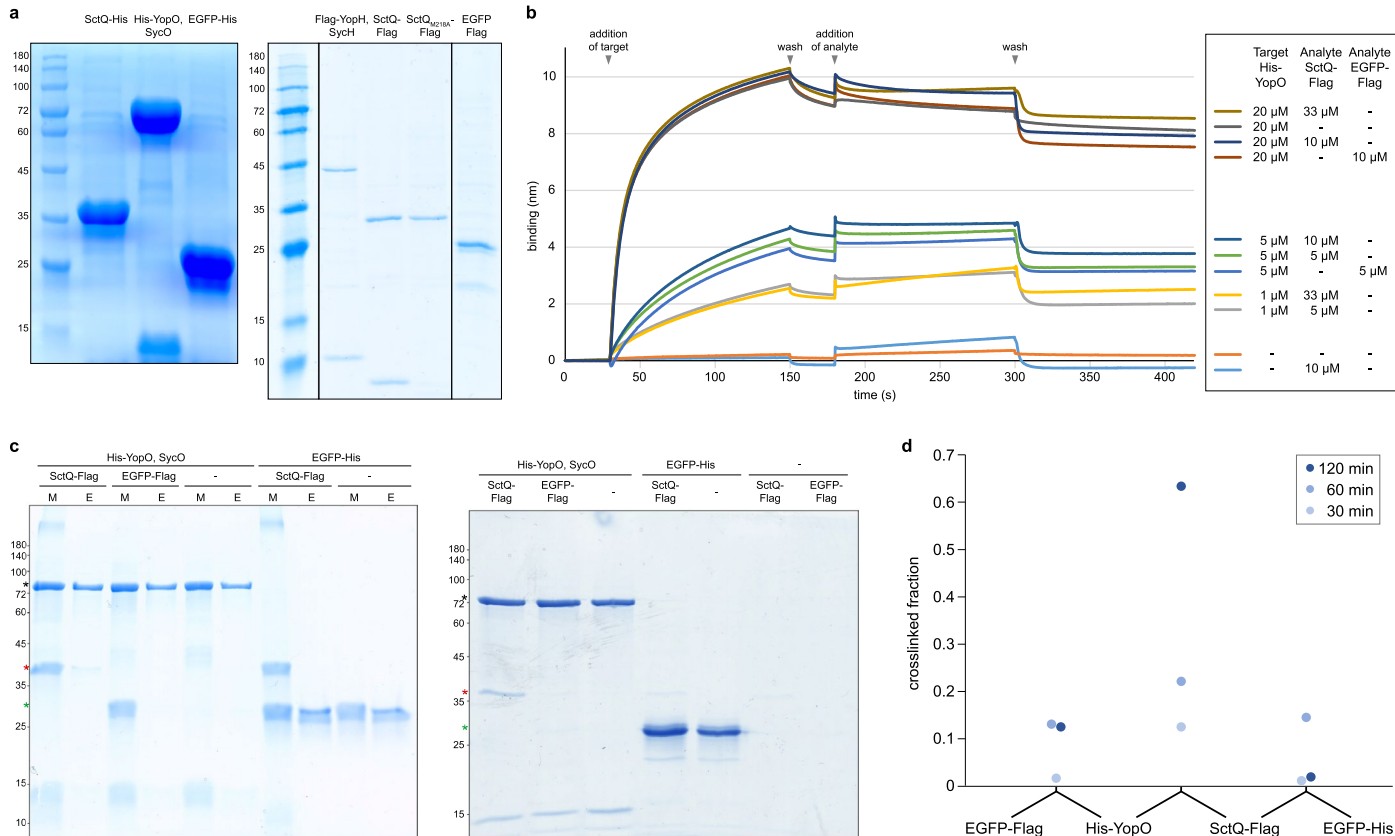

**Extended Data Fig. 2 | *In vitro* interaction analysis of purified SctQ and YopO proteins by crosslinking-purification and biolayer interferometry.**
**a**) Overview of purified proteins, as indicated, used in the interaction assays. SctQ$_{M218A}$-Flag was used to test for the copurification of SctQ$_C$ in the SctQ-Flag purification. Expected sizes from top to bottom: left, His$_8$-YopO, 82.7 kDa; SctQ-His$_8$, 35.9 kDa; EGFP-His$_8$, 28.5 kDa, copurified SycO, 17.2 kDa; right, Flag-YopH, 52.3 kDa; SctQ-Flag, 35.8 kDa; EGFP-Flag, 28.4 kDa; copurified SycH, 15.7 kDa; copurified SctQ$_C$, 10.0 kDa). Note that highly concentrated or acidic proteins, such as the T3SS chaperones, may migrate faster than expected on SDS-PAGE gels[1], compare to panel c for migration at lower concentration. **b**) Representative biolayer interferometry binding sensorgram showing the affinity of SctQ-FLAG towards the target His-YopO (bait protein) immobilized on Ni-NTA biosensor tip. Varying concentrations of His-YopO, as indicated, were loaded on the tips and exposed to different concentrations of SctQ-FLAG for 120 s (association phase),

followed by a 120 s washing step (dissociation phase). The y-axis displays the shift in light wavelength upon binding measured in nanometers (nm) as a function of time (seconds). Purified EGFP-FLAG was used as a control. $n = 3$. **c**) Left, crosslinking-purification assay of the indicated protein pairs. M, input (mixture of the indicated proteins); E, eluate after Ni$^{2+}$-NTA-based purification of proteins after 30 min of crosslinking (see Methods for details). Asterisks denote expected proteins sizes (from top to bottom, His$_8$-YopO, 82.7 kDa; SctQ-Flag, 35.8 kDa; EGFP-Flag, 28.4 kDa). Right, direct comparison of eluate samples from same experiment including controls without His-tagged protein, 2x volume loaded onto SDS-PAGE gel; asterisks as indicated earlier. **d**) Quantification of interactions by crosslinking. The indicated protein pairs were incubated with crosslinker for the indicated periods of time; interaction was quantified by comparing the intensity of the respective Flag-tagged protein before and after crosslinking on immunoblots using anti-Flag antibodies. $n = 3$.

**a** live cells

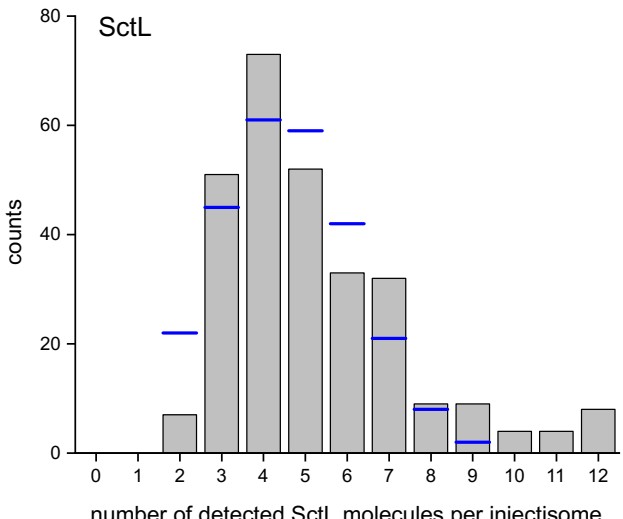
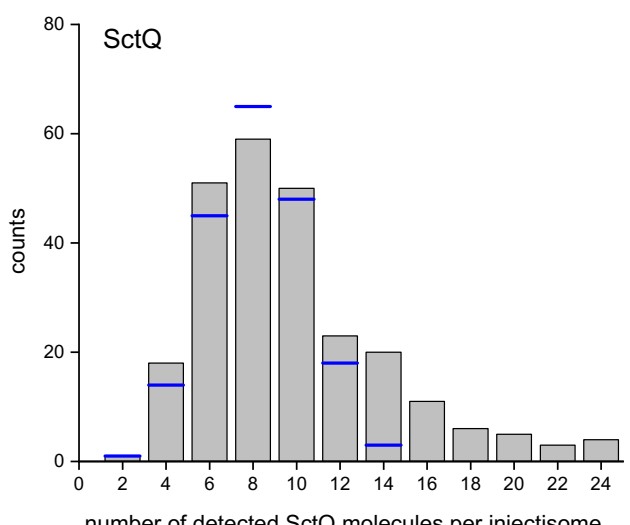

**b** fixed cells

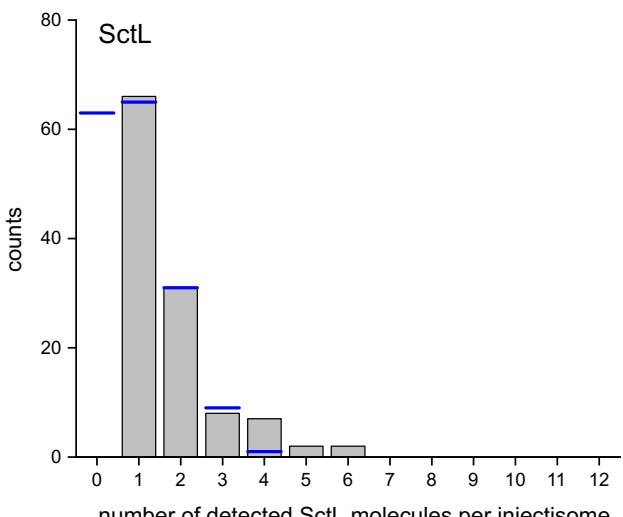
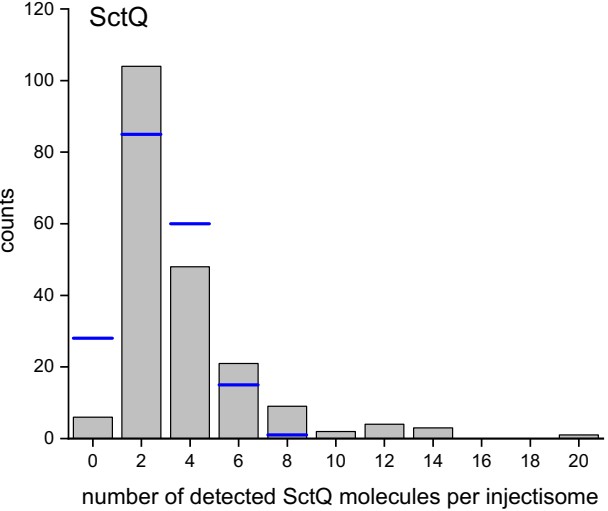

**Extended Data Fig. 3 | Determination of the stoichiometry of SctQ and SctL per injectisome.** Quantification of membrane-bound immobile PAmCherry-SctL and PAmCherry-SctQ molecules per injectisome detected by sptPALM, strongly indicating the presence of 24 SctQ subunits per injectisome. SctL with a stoichiometry of 12 subunits[6–10] was used as a reference to determine the stoichiometry of SctQ subunits per injectisome. Histograms of detected SctL or SctQ molecules per injectisome measured in live cells **(a)** and fixed cells **(b)**. Blue lines represent the corresponding best binomial fit obtained using a custom-written script in Python (v.3.8.5). **a)** Live cells: For PAmCherry-SctL, a median of 5 detected molecules per injectisome was determined (mean: $5.2 \pm 2.2$); for PAmCherry-SctQ, a median of 9 molecules per injectisome (mean: $10.3 \pm 4.3$). This is highly compatible with a 1:2 stoichiometry of SctL:SctQ. Accordingly, when comparing the binomial fits of SctL with $N = 12$ molecules per injectisome to the binomial fit of SctQ with $N = 24$, the detection probabilities $\bar{p}$ for SctL and SctQ are very similar, ($\bar{p} = 0.44$ and $\bar{p} = 0.43$, respectively). This is expected as the same fluorophore and imaging protocol are used for the read-out of both molecules. Thus, the detection probability should not change between SctL and

SctQ samples. Furthermore, statistical tests confirm a high goodness of fits with $r^2 = 0.89$ (SctL, $N = 12$) and $r^2 = 0.91$ (SctQ, $N = 24$). In contrast, binominal fits assuming $N = 12$ or $N = 6$ SctQ subunits per injectisome result in highly dissimilar detection probabilities and worse fit to the data with $\bar{p} = 0.86$, $r^2 = 0.78$ ($N = 12$), and $\bar{p} = 1.72$, $r^2 = 0.04$ ($N = 6$). **b)** Fixed cells: A median of 1 detected PAmCherry-SctL molecule per injectisome (mean: $1.9 \pm 1.7$) and a median of 3 detected PAmCherry-SctQ molecules per injectisome (mean: $4.2 \pm 2.8$) were detected, again compatible with a 1:2 stoichiometry of SctL:SctQ. The decreased number of detected molecules compared to the live cell data in a) can be explained by altered photophysics and lower chromophore retention of PAmCherry in fixed cells, which leads to less efficient detection[11–13]. Similar to the live cell data in a), a stoichiometry of $N = 24$ for SctQ yields very similar detection probabilities, compared to the $N = 12$ reference for SctL, and good fits to the data ($\bar{p} = 0.16$ and $\bar{p} = 0.17$, $r^2 = 0.99$ and $r^2 = 0.94$, for SctL and SctQ, respectively), whereas $N = 12$ or $N = 6$ for SctQ result in highly different detection probabilities and worse fits to the data ($\bar{p} = 0.35$, $r^2 = 0.94$ for $N = 12$; $\bar{p} = 0.69$, $r^2 = 0.62$ for $N = 6$).

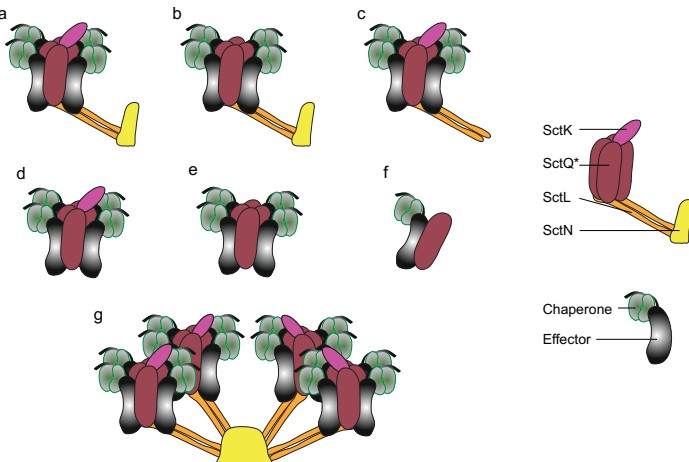

**Extended Data Fig. 4 | Sorting platform subcomplex compositions.** Depiction of sorting platform proteins and subcomplexes binding to effectors, as used for calculations of predicted mobility values for the mean jump distance in Table 2.

Letters refer to subcomplexes listed in Table 2: **a**) Single pod; **b**) Single pod lacking SctK; **c**) Single pod lacking SctN; **d**) SctK(Q(Q$_C$)$_2$)$_4$; **e**) (SctQ(Q$_C$)$_2$)$_4$; **f**) SctQ(Q$_C$)$_2$; **g**) Complete sorting platform. *, SctQ depicts SctQ(Q$_C$)$_2$ subcomplex.

# Reporting Summary

## Statistics

For all statistical analyses, confirm that the following items are present in the figure legend, table legend, main text, or Methods section.

| n/a | Confirmed | |
|---|---|---|
| ☐ | ☒ | The exact sample size (*n*) for each experimental group/condition, given as a discrete number and unit of measurement |
| ☐ | ☒ | A statement on whether measurements were taken from distinct samples or whether the same sample was measured repeatedly |
| ☐ | ☒ | The statistical test(s) used AND whether they are one- or two-sided *Only common tests should be described solely by name; describe more complex techniques in the Methods section.* |
| ☒ | ☐ | A description of all covariates tested |
| ☒ | ☐ | A description of any assumptions or corrections, such as tests of normality and adjustment for multiple comparisons |
| ☐ | ☒ | A full description of the statistical parameters including central tendency (e.g. means) or other basic estimates (e.g. regression coefficient) AND variation (e.g. standard deviation) or associated estimates of uncertainty (e.g. confidence intervals) |
| ☐ | ☒ | For null hypothesis testing, the test statistic (e.g. *F*, *t*, *r*) with confidence intervals, effect sizes, degrees of freedom and *P* value noted *Give P values as exact values whenever suitable.* |
| ☒ | ☐ | For Bayesian analysis, information on the choice of priors and Markov chain Monte Carlo settings |
| ☒ | ☐ | For hierarchical and complex designs, identification of the appropriate level for tests and full reporting of outcomes |
| ☒ | ☐ | Estimates of effect sizes (e.g. Cohen's *d*, Pearson's *r*), indicating how they were calculated |

*Our web collection on statistics for biologists contains articles on many of the points above.*

## Software and code

Policy information about availability of computer code

| Data collection | sptPALM images were acquired on a Nikon Ti Eclipse microscope, using the Micro-Manager software version 1.4.23. |
|---|---|
| Data analysis | Rapidstorm software version 3.3.1 was used for single-molecule localization. ImageJ 1.51v./1.52p/1.53c based Fiji software package was used for cell segmentation. Custom-written tracking software Swift was used for tracking, visualizing and filtering sptPALM data (http://bit.ly/swifttracking; version 0.3.1, used in this manuscript, and all subsequent versions of the software can also be obtained upon request to the authors). Binomial distribution fit analysis was performed with Python version 3.8.10 using the SciPy optimize library. Plotting of data was performed using Origin 2019. Protein data was analyzed with MaxQuant in standard settings (Tyanova et al, 2016). If further statistical analysis was required, the MaxQuant "proteinGroups.txt" file was further evaluated using SafeQuant 2.4 (Ahrné et al, 2016, https://github.com/georgiaAngelidou/SafeQuant.v2.3.5, current versions available upon request to the authors). Standard spreadsheet calculations were performed using Microsoft Excel 2016. |

For manuscripts utilizing custom algorithms or software that are central to the research but not yet described in published literature, software must be made available to editors and reviewers. We strongly encourage code deposition in a community repository (e.g. GitHub). See the Nature Portfolio guidelines for submitting code & software for further information.

## Data

Policy information about availability of data

All manuscripts must include a data availability statement. This statement should provide the following information, where applicable:
- Accession codes, unique identifiers, or web links for publicly available datasets
- A description of any restrictions on data availability
- For clinical datasets or third party data, please ensure that the statement adheres to our policy

All relevant data are included in the paper and/or its Supplementary information files. Source data are provided with this paper. The mass spectrometry proteomics data have been deposited to the ProteomeXchange Consortium via the PRIDE 104 partner repository with the dataset identifier PXD044214 (see Suppl. Table 6 for assignment). Supplementary videos are accessible at https://doi.org/10.17617/3.HMABQ2

## Human research participants

Policy information about studies involving human research participants and Sex and Gender in Research.

| | |
|---|---|
| Reporting on sex and gender | Not applicable. |
| Population characteristics | N/A |
| Recruitment | N/A |
| Ethics oversight | N/A |

Note that full information on the approval of the study protocol must also be provided in the manuscript.

# Field-specific reporting

Please select the one below that is the best fit for your research. If you are not sure, read the appropriate sections before making your selection.

☒ Life sciences          ☐ Behavioural & social sciences          ☐ Ecological, evolutionary & environmental sciences

For a reference copy of the document with all sections, see nature.com/documents/nr-reporting-summary-flat.pdf

# Life sciences study design

All studies must disclose on these points even when the disclosure is negative.

| | |
|---|---|
| Sample size | No preliminary sample-size calculation was determined. Samples sizes were chosen to allow a clear test of the conclusions drawn from the presented data based on similar experiments, or as n=3 where no such previous information was available. As the variance across experiments was low in most case, no explicit power calculations were considered necessary. |
| Data exclusions | No data were excluded. |
| Replication | Experiments were performed in independent replicates as indicated in the respective figure legends, material and methods, and/or the Supplementary Information, specifically Suppl. Table 2. |
| Randomization | Random allocation with regard to covariate is not applicable to the experiments carried out; no allocation into experimental groups was performed. |
| Blinding | Blinding was not required as the data processing did not rely on subjective assignments. |

# Reporting for specific materials, systems and methods

We require information from authors about some types of materials, experimental systems and methods used in many studies. Here, indicate whether each material, system or method listed is relevant to your study. If you are not sure if a list item applies to your research, read the appropriate section before selecting a response.

## Materials & experimental systems

| n/a | Involved in the study |
|---|---|
| ☐ | ☒ Antibodies |
| ☒ | ☐ Eukaryotic cell lines |
| ☒ | ☐ Palaeontology and archaeology |
| ☒ | ☐ Animals and other organisms |
| ☒ | ☐ Clinical data |
| ☒ | ☐ Dual use research of concern |

## Methods

| n/a | Involved in the study |
|---|---|
| ☒ | ☐ ChIP-seq |
| ☒ | ☐ Flow cytometry |
| ☒ | ☐ MRI-based neuroimaging |

## Antibodies

| | |
|---|---|
| Antibodies used | Primary antibodies: rabbit antibodies against mCherry (Biovision 5993, 1:2000) or against the Flag peptide (Rockland 600-401-383S, 1:5000)<br>Secondary antibodies: anti-rabbit antibodies conjugated to horseradish peroxidase (HRP) (Sigma A8275, 1:5000) |
| Validation | Validation statements and examples on manufacturers' sites: https://www.abcam.com/mcherry-antibody-ab286186.html, https://www.rockland.com/categories/primary-antibodies/antibody-for-the-detection-of-flag-conjugated-proteins-600-401-383/, https://www.sigmaaldrich.com/GB/en/product/sigma/a8275. Additional validation by controls used in respective experiments. |

