## [Peer Review File · Nature Microbiology]

Peer Review Information

Journal: Nature Microbiology

Manuscript Title: Cytosolic sorting platform complexes shuttle type III secretion system effectors to the injectisome in *Yersinia enterocolitica*

Corresponding author name(s): Dr Andreas Diepold, Professor Ulrike Endesfelder

Reviewer Comments & Decisions:Decision Letter, initial version:**Message:** 6th October 2022

Dear Professor Diepold,

Thank you for submitting your Article entitled "Direct binding of type III secretion effectors to cytosolic sorting platform complexes in live bacteria indicates an effector shuttling mechanism" for consideration in Nature Microbiology and please accept our apologies for the time it has taken us to contact you with a decision on your manuscript, which is due to our current high submission volume. I regret to inform you that after careful discussion within the editorial team, we have decided that we cannot consider it for publication here.

As you may know, we decline a substantial proportion of manuscripts without sending them to referees, so that they may be sent elsewhere without delay. In such cases, even if referees were to certify the manuscript as technically correct, we consider that the work does not represent the type of advance that Nature Microbiology seeks to publish. This editorial assessment is based on considerations such as the degree of conceptual advance provided, the breadth of potential interest to researchers and timeliness.

In this case, although we have no doubt of the interest that your study will have to others working in this field, in terms of the overall degree of advance and immediate relevance to our broad microbiological readership, we do not feel that the current work has quite met the high bar that we must unfortunately set for further consideration towards publication in Nature Microbiology. We therefore feel that the paper would find a more suitable outlet in another journal.

Although we cannot offer to publish your manuscript, I suggest that you consider transferring your manuscript to the Springer Nature journal *Nature Communications*, which is the Nature Research flagship Open Access journal. If you would like this work to be considered for publication there, you can easily transfer the manuscript by following the instructions below. It is not necessary to reformat your paper.

Please be assured that this editorial decision does not represent a criticism of the quality of your work, nor are we questioning its value to others working in this area. We hope that you will rapidly receive a more favourable response elsewhere.

I am sorry that we cannot respond more positively on this occasion.

Yours sincerely,
[redacted]

Author Rebuttal to Initial comments

Dear [redacted],

Thank you for your mail and assessment of our manuscript. Reading your reply, we feel that we sadly didn't highlight the context and importance of our findings well enough and so would like to point them out more clearly in this short response to your points raised, especially concerning the perceived lack of conceptual advance, breadth of potential interest, and timeliness.

The central finding of the manuscript, the **binding of effectors to the T3SS sorting platform in the cytosol** of live bacteria and the **direct evidence in support of an effector shuttle model**, transforms our understanding of the recruitment of effectors to the T3SS. This is one of the **central questions in the field**, as uniformly stated in recent reviews by leading scientists in the field:

Deng, (...), Strynadka & Finlay (2017), 10.1038/nrmicro.2017.20: "To move the field forward, there should be attempts to fully elucidate the molecular mechanisms of the control of secretion hierarchy and the recognition of secretion signals, and determine whether there could be a unified model for the control of needle length."

Wagner *et al.* (2018), 10.1093/femsle/fny20: "Unfortunately, however, our understanding of the molecular mechanisms of substrate targeting and secretion, of needle-length control and substrate specificity switching, of host-cell sensing and translocon assembly, is still rudimentary and awaits detailed biochemical investigation."

Lara-Tejero & Galán (2019), 10.1128/ecosalplus.esp-0039-2018: "Although there have been remarkable advancements in the understanding of the structure and function of these machines, many knowledge gaps still remain. For example, atomic information on some essential elements of the injectisome is still missing, the precise mechanisms of substrate engagement by the secretion machine are poorly understood, and the mechanisms of eukaryotic cell sensing and signal transduction remain obscure. Addressing some of these fundamental issues most likely will require the development of novel experimental approaches to be able to study the function of these machines in live bacteria"

Max Planck Institute for Terrestrial Microbiology

Karl-von-Frisch Straße 10

Tel.: +49 6421 178-0

35043 Marburg Germany

www.mpi-marburg.mpg.de

In addition to addressing these key questions, our work clarifies the **role of the sorting platform**, a concept which, despite its popularity (>280 citations of the 2011 Science paper that introduced the “sorting platform” concept and more than a thousand mentions in publications since 2017 alone) has been founded on a very thin data base.

We therefore believe that the mechanistic and conceptual advance presented in our manuscript is not only of great interest to researchers working in the highly active field of type III secretion, but also to the wide audience of researchers interested in bacterial protein export and host-pathogen interactions, especially involved in human health. Discussing the fit of our work for *Nature Microbiology* among the authors, we actually realized that this research field has so far been underrepresented in *Nature Microbiology* and are wondering why.

From a technical point of view, we for the first time can showcase that a single particle tracking-based “***in vivo* biochemistry**” approach is a method that can detect highly transient, cytosolic interactions, i.e. the sorting platform dynamics, that cannot be measured by other technologies. Our work thus has the strong potential to advance diverse areas of microbiology and hopefully will serve as a blueprint to many.

To summarize, we hope that based on this additional information, you may reconsider our manuscript and send it out for peer review. For the likely case that our letter will not change the outcome of the first assessment as common for most rebuttals, we would be very grateful to learn and understand more about the basis of your assessment and whether it was based on the lack of potential interest in bacterial protein secretion of the journal as such or on specific aspects, both biological or technical, that are missing in our manuscript. We would strongly appreciate this feedback – thanks a lot!

On behalf of all authors, thank you and best wishes,

Andreas Diepold

Decision Letter, first revision:

Message: 5th December 2022

*Please ensure you delete the link to your author homepage in this e-mail if you wish to forward it

1to your co-authors.

Dear Andreas,

Thank you for your patience while your manuscript "Direct binding of type III secretion effectors to cytosolic sorting platform complexes in live bacteria indicates an effector shuttling mechanism" was under peer-review at Nature Microbiology. It has now been seen by 3 referees, whose expertise and comments you will find at the end of this email. Although they find your work of some potential interest, they have raised a number of concerns that will need to be addressed before we can consider publication of the work in Nature Microbiology.

In particular, referee #1 has some serious concerns that what is described in the manuscript may be an artifact, and that the data to support the conclusions is insufficient. For example, the referee mentions that the study does not provide sufficient evidence for interactions between C-ring components and effector proteins, and that an essential control missing in this study is the use of a YscQ mutant that is unable to bind effectors but retains the ability to bind sorting platform partners. Further, this referee says that no evidence is presented that indeed the EscQ/EscL complex co-localize with effectors in the cytoplasm. Referee #1 also says that the conclusions are overstated and the methods insufficiently well described. Referee #2 shares some of the concerns of referee #1, and feels that some of the claims are not supported by the data. They also feel that some conclusions would need to be validated with other experimental approaches, for example, they are not convinced that the mass spec data are suited to claim interaction strength of proteins. Referee #3 also says the methods are not well described, and the whole paper would need thorough editing for language and accessibility.

Should further experimental data allow you to address these criticisms in full, we would be happy to look at a revised manuscript.

We strongly suggest that you first prepare a short revision plan describing how you plan to experimentally address the referees' concerns, especially those of referee #1. Please send this revision plan to me via email. We will then discuss it within the team and then get back to you with some feedback.

Please include a data availability statement as a separate section after Methods but before references, under the heading "Data Availability". This section should inform readers about the availability of the data used to support the conclusions of your study. This information includes accession codes to public repositories (data banks for protein, DNA or RNA sequences, microarray, proteomics data etc...), references to source data published alongside the paper, unique identifiers such as URLs to data repository entries, or data set DOIs, and any other statement about data availability. At a minimum, you should include the following statement: "The data that support the findings of this study are available from the corresponding author upon request", mentioning any restrictions on availability. If DOIs are provided, we also strongly encourage including these in the Reference list (authors, title, publisher (repository name), identifier, year). For more guidance on

how to write this section please see:

<http://www.nature.com/authors/policies/data/data-availability-statements-data-citations.pdf>

- * Include a "Response to referees" document detailing, point-by-point, how you addressed each referee comment. If no action was taken to address a point, you must provide a compelling argument. This response will be sent back to the referees along with the revised manuscript.
- * If you have not done so already we suggest that you begin to revise your manuscript so that it conforms to our Article format instructions at <http://www.nature.com/nmicrobiol/info/final-submission>. Refer also to any guidelines provided in this letter.
- * Include a revised version of any required reporting checklist. It will be available to referees (and, potentially, statisticians) to aid in their evaluation if the manuscript goes back for peer review. A revised checklist is essential for re-review of the paper.

When submitting the revised version of your manuscript, please pay close attention to our [href="https://www.nature.com/nature-portfolio/editorial-policies/image-integrity">Digital Image Integrity Guidelines.](https://www.nature.com/nature-portfolio/editorial-policies/image-integrity) and to the following points below:

[redacted]

Note: This url links to your confidential homepage and associated information about manuscripts you may have submitted or be reviewing for us. If you wish to forward this e-mail to co-authors, please delete this link to your homepage first.

Nature Microbiology is committed to improving transparency in authorship. As part of our efforts in this direction, we are now requesting that all authors identified as 'corresponding author' on published papers create and link their Open Researcher and Contributor Identifier (ORCID) with their account on the Manuscript Tracking System (MTS), prior to acceptance. This applies to primary research papers only. ORCID helps the scientific community achieve unambiguous attribution of all scholarly contributions. You can create and link your ORCID from the home page of the MTS by clicking on 'Modify my Springer Nature account'. For more information please visit please visit www.springernature.com/orcid.

If you wish to submit a suitably revised manuscript we would hope to receive it within 6 months. If you cannot send it within this time, please let us know. We will be happy to consider your revision, even if a similar study has been accepted for publication at Nature Microbiology or

published elsewhere (up to a maximum of 6 months).

Yours sincerely,
[redacted]

Reviewer Expertise:

Referee #1: Single particle tracking super-resolution microscopy, type III secretion systems

Referee #2: Superresolution imaging and single particle tracking in bacteria

Referee #3: Type III secretion systems

Reviewer Comments:

Reviewer #1 (Remarks to the Author):

In this paper, Diepold and collaborators examined mechanisms by which type III secreted client proteins may be recruited to the secretion machine. Type III secretion systems (T3SSs) are composed of a bacterial-envelope-embedded structure known as the "basal body", and a cytoplasmic complex known as the C-ring (also known as the "sorting platform"). Cryo-ET studies have shown that the C-ring or sorting platform is composed of 6 pods, each one made of the SctK, SctL, and SctQ proteins. This structure provides a scaffold for a hexameric ATPase (SctN) to place it in close register to an enclosed nonameric ring structure composed of a single protein, SctV, which acts as the "export gate" and receptor for client proteins. The mechanisms by which translocases and effector proteins are recruited to the secretion machine are incompletely understood. The current model supported by a large amount of data states that effector-chaperone complexes reach the sorting platform by diffusion, bind to "receptor" proteins (most likely SctV, SctN, or both), and are initiated into the secretion pathway. Different affinities of the different client proteins for the receptors dictates the order in which they are secreted. This model accommodates the available data and the rather rapid secretion rate that has been measured in several systems. Diepold and collaborators, on the other hand, have proposed an alternative model in which the entire sorting platform assembles and disassembles during secretion, a process that, as expanded in this study, results in the recruitment of client proteins to the secretion machine after their shuttling by the components of sorting platform themselves. This model was originally advanced based on diffraction-limited FRAP studies of a component of the sorting platform (SctQ) of the T3SS of the bacterial pathogen *Yersinia* but has not been corroborated for any other T3SS or by any other laboratory. The T3SS community has been sceptic about this model because of the "soft" nature of the original data (diffraction-limited FRAP), and its rather counter intuitive nature (i. e. how the cycle of assembly/disassembly of the C-ring could possibly keep up with the fast rate of secretion). Furthermore, visualization of the T3SS structure by cryo ET has shown the presence of the sorting platform in almost 100% of the secretion machines, an observation that is in principle incompatible with the notion of continuous assembly/disassembly during its activity. These observations provide a context for the study described in this paper, where it is proposed that components of the C-ring (SctQ/SctL) shuttle effector proteins to the secretion machine. This conclusion is reached with remarkably little data to support it (see below). In addition, there are serious experimental shortcomings that raises question about the reliability of the imaging data obtained. Given the rather contested nature of the original claim, premature publication of these results without the appropriate evidence to support it would only add to the skepticism and would not serve well the authors or the T3SS research community at large.

Specific comments:

1) The entire premise that SctQ presumably working in conjunction with SctL act as a shuttle for the effector/chaperone complexes recruiting them to the C ring or sorting platform is supported by observations that the mobility of SctQ and SctL, as measured by single particle tracking PALM (sptPALM) is "slowed down" in the presence of effector proteins. It is assumed that such reduction in the diffusion of SctQ and SctL is due to being bound to the effector/chaperone complex. However, no direct evidence is provided to support this premise. This is a major shortcoming as, even if their measurements were to be correct (although see below), there are alternative explanations for their observations that cannot be ruled out with the data presented. In fact, the authors attempted to demonstrate direct-binding but they failed. They ascribe their failure to a proposed "transient binding" nature of the interaction. However, they only attempted to measure interactions by co-immunoprecipitation in the presence of detergents, which is a rather crude method to attempt to measure protein-protein interactions. There are multitude of experimental approaches that the investigators could have deployed to detect even the most transient of interactions. In the context of the failure of the T3SS community to detect interactions between C-ring components and effector proteins with even more sensitive approaches, without this crucial piece of evidence, the authors cannot draw the conclusions that they are drawing.

2) An essential control missing in this study is the use of a YscQ mutant that is unable to bind effectors but retains the ability to bind sorting platform partners. This reviewer understands that without knowing whether YscQ even binds to client proteins to begin with, identifying a mutant may be challenging but to advance the hypothesis the authors are advancing with the experimental tools they are using, such a control is essential.

3) Surprisingly, no evidence is presented that indeed the EscQ/EscL complex co-localize with effectors in the cytoplasm as their model would predict; this is a crucial piece of data also missing from this study.

4) A unique aspect of the Yersinia T3SS is that its expression, assembly, and activation can be induced by specific growth conditions (used in this study; 37 °C in the absence of Ca). It should be stated that these conditions, while experimentally useful, are highly artificial so much so that it leads to bacterial growth arrest, a phenotype original used by Brubaker, Straley, Goguen and others to identify the Yersinia T3SS genes in this bacterium (the so called Lcr or "low Ca response" genes). Since expression, assembly, and secretion are going on at the same time during the experimental conditions used in these studies, the authors cannot rule out that a proportion (or the totality) of the "movement" of SctQ/SctL detected by sptPALM may be related to the journey from their place of synthesis in the cytoplasm to their destination to form the C ring. Even if the measurements were to be correct and SctQ's mobility in the cytoplasm is reduced in the presence of effectors, the authors cannot rule out an artifact stemming from YscQ being bound prematurely in the cytoplasm by the effectors prior to its recruitment to the C ring because the conditions used in the experiment are such that multiple events are going on at the same time: T3SS expression, injectisome assembly, and client protein secretion. Experiments must be done to uncouple these events. In other words, experiments should be done in which the injectisome is assembled first, and then effectors are expressed without de novo synthesis of YscQ. Otherwise, the multiple events cannot be untangled. The rather artificial nature of the Yersinia system has led in the past to many artifactual observations that when observed in other systems, have not held up...this may well be another case..

5) In the absence of basal body components, it has been shown that EstQ aggregates, which can of course affect its mobility. This possibility has not been ruled out in these studies. Also, overexpression of SctQ can lead to a proportion of molecules misfolded, which would also alter their mobility.

6) The data does not address the need for a shuttling protein that will "transport" effectors to the sorting platform when diffusion alone would just do the same more efficiently!! Effectors could then be simply "trap" by the sorting platform. In other words, diffusion will always outpace even

the fastest secretion machine. The model proposed is not only counter-intuitive but inconsistent with the rapid speed of secretion.

7) The mobility measurements have been done in 2D. However, diffusion occurs in 3D. Consequently, the actual rates measured do not reflect reality. Measurements should be done in 3D. It is unclear why the authors have chosen to do them in 2D.

8) The methodology is very poorly described and with the information given, it would be impossible, even for a laboratory with a lot of experience in high-end imaging to reproduce these studies. For example, for some experiments the authors state that they have eliminated information related to molecules present at the membrane. Yet, the way the experiments were conducted, it would be impossible to identify pixels at the membrane since bacteria are 3D objects and the way the images appear to have been collected, it would be impossible to distinguish molecules in the cytoplasm from those at the membrane on top or below. It should be said that sptPALM was originally described by Lippincott-Schwartz and Betzig to investigate movement of molecules at the membrane, which is a 2D environment and not a 3D environment like the one being imaged in these studies. Also, it is unclear whether only 1 jump was measured or multiple jumps. In that case, new molecules being photo-activated would interfere with the interpretation of the results.

9) The conclusions drawn related to the composition/stoichiometry of a single pod are highly speculative and are not supported by the data. No conclusions can be drawn as to the approach on stoichiometry of complexes with the approach used in these studies. Inferences made from the values stated in Table 1 cannot possibly be correct since the authors have no information about the real composition or shape of the complexes. Yet, an inexperienced reader may get the idea that those values reflect some reality, which it is not the case. The authors do not help by making aggressive claims not supported by the data.

7) The Δ effector strain used in some of the studies still retain the protein translocases. Yet, the authors observed an apparent difference in the mobility of SctQ in comparison with wild type, a difference that is equivalent to what they observed in the absence of the plasmid (i. e. in the absence of translocases and effectors). This makes little sense even in the context of the model presented by the authors since the translocases would also bind SctQ and slow it down. Secretion of the protein translocases is equally dependent on the C ring components and there are no data indicating that their engagement by the secretion machine occurs by different mechanisms. These observations cast serious doubts about what the authors are actually measuring or what does it all mean...

8) If binding of SctQ to the effectors is so transient, how do the authors consider into their trajectories the kinetics of unbinding?

9) How do the authors explain the much slower mobility of SctL (25kDa) compared to the bigger proteins SctQ (37 kDa) and SctN (47 kDa)? Even if one takes a look at Table 1, the mobility peak of PAmCherry-SctL with no effectors (~300 nm) is comparable to much bigger "predicted" complexes. Makes little sense...

Reviewer #2 (Remarks to the Author):

Type III secretion systems are widespread in bacteria and they are used to inject a variety of effector proteins into eukaryotic cells. The structure and composition of T3SS is conserved and also the flagella apparatus shares the conserved structure and functions. In recent years a wealth of new data on the structure and assembly of T3SS has been published. However, the assembly

dynamics, as in most cases of large cellular complexes, remained insufficient understood. In particular interaction of the secreted substrates with the cytosolic sorting platform is unclear and under discussion. The authors use here single particle tracking (sptPALM) to unravel dynamics of sorting platform proteins and substrates in presence and absence of components in live *Yersinia enterocolitica* cells. They complement their analyses with co-immunoprecipitation and mass spectrometry data. Overall, this work is a nice contribution to the field and the experiments are carefully executed and written up in a concise and coherent manuscript. In particular, the SPT data are very impressive and of high quality. A weak point is that the major claims are based on assumptions and calculations of theoretical diffusion of proteins and complexes. While I understand that there are structural data and other biochemical data that suggest existence of certain sorting platform sub complexes, the data are rather indirect. Below I have summarized a couple of points that might be helpful to revise the manuscript.

Main points:

Supplementary Figures 1, 2 and 4 show immunoblot data of PAmCherry labelled T3SS proteins. I am puzzled by the differences in stability. PA-mCherry-SctQ in Fig. S1 appears reasonable stable as a full length band, migrating above the 72 kDa marker. In Fig. S2a the same construct appears well below the 72 kDa marker (red asterisk). In line 1 there is barely any full length protein visible, while a clear degradation band is prominent. The elution fraction shows two major bands. Although there seems no free PA-mCherry, the existence of degradation products will complicate the SPT analysis. Quite similar is the problem for SctL and SctN (Fig. S2B-C).

Supplementary Fig. 3 shows the data of mass spectrometry analysis with SctQ, SctL and SctN as bait proteins. I did not find any hint of where the mass spec raw data were deposited in a database. While the interaction of the sorting platform proteins with substrate proteins (Yop) is clear (also from previous publications), I am not sure how these data are in agreement with the final model. There the Yop proteins are indicated to bind to SctQ only, while the mass spec data also suggest interaction with SctLN in some cases. Using Mass spec data to quantify interaction strength is in my opinion to much speculation. Protein-protein interactions can quantitatively be addressed by various methods including SPR, BLI, ITC etc., but that would require more in vitro work. I therefore suggest to discuss this better in the text and tone down the statements.

The dynamic behavior of the T3SS proteins is estimated based on a jump distance analysis. This is a very valid procedure, but it does not allow comparison with other protein diffusion data. I assume the authors relied on Jump distance analysis because all other calculations of diffusion coefficients are error prone. Still I was wondering if there is a chance at all to compare diffusion data between different experiments to allow a comparison between the new and published data on other complexes. In particular, since the authors used these coefficients in their calculations. This could maybe clearer in the text.

I agree that transient interactions of proteins and complexes are difficult to capture in live cells. However, there are several biochemical and genetic ways to overcome these limitations. Cross-linking by introducing cysteine residues at critical sites has been extensively used to stabilize transient interaction. It also offers a nice confirmation of structural organizations. This is of course involving a lot of experimental effort. An easier, but less precise way is crosslinking with fixatives.

Would the high resolution of SMLM be sufficient to gain insight into the structure of the T3SS sorting platform and the interaction with the injectisome? Ideally, using dual color SMLM could bring by additional information on spatial arrangements.

Minor points

Line 164: What do the authors mean with "almost" no effectors?

Line 271: remove "earlier"

Figure 3 legend: line 295 - the last band is labelled "green" not blue as in the text.

Line 324: Therefore, we in situ analyzed - rephrase

Lines 388: This work is not the first use of SPT to study protein interaction and dynamics in vivo.

This may be toned down.

Line 427: The authors speak about "rearrangements", this should be better explained.

Line 510: Quantification of foci: The text reads as if all stable foci that are not at the outline of the cell were not shown. How many of these clusters were observed? If they are there, what are they?

Reviewer #3 (Remarks to the Author):

The significance of this work is that it takes us from conjecture that the sorting platform shuttles effector proteins to the injectisome to the actual data to support the hypothesis that this actually occurs (and quantitatively no less!). The authors combine various experimental approaches to provide important insights into the initiation of the secretion process and sequestration of effectors prior to their secretion and translocation into target host cells. The first chapter describes interactions between PAmCherry-SctQ PAmCherry-SctL and PAmCherry-SctN and the YopO/SycO effectors. The data is convincing for interactions with SctQ and SctL and not SctN, but the authors do not describe the methodology very well for a non-expert. Chapter 2 analyzes the mobility of sorting platform under secreting and on-secreting conditions using the PAmCherry-SctQ fusion and YopO/SycO substrates by the same method. The data is solid and clearly show ratio of effector/SctQ increases significantly compared to effector-free condition from non to secreting bacteria. Finally, they show binding of effectors to the sorting platform in live bacteria in the cytosol followed by convincing experiments that SctK&L, while apart of the sorting platform are not required to bind effectors, again in live individual cells.

This is actually a very cool and important study with very cutting-edge methodology employed. Please make the writing what it should be for such quality science: a pleasure to read.

I made minor comments through line 155, but stopped for lack of time. This needs editing, preferable by someone not in the field so that the targeted average reader of this Journal, a broad-audience journal, can understand the science, which first requires that the numerous grammatical problems be addressed.

Abstract

Line 2: change "The bacterial type III secretion system, also called the injectisome, translocates" to "Bacterial type III secretion is utilized by injectisome structures to translocate". The bacterial type III secretion system is not called the injectisome. Both injectisomes and flagella utilize type III secretion to secrete proteins.

Lines 7-8: this sentence makes no sense. I believe you are trying to say "There are ~5 injectisomes per cell under non-secreting growth conditions and ~18/cell under secreting growth conditions."

Lines 8-9: another non-comprehensible sentence. Who are "they"? A mobile cytosolic pool of what? There are eleven authors on this manuscript, does anyone understand English grammar? You have to understand that, as a reviewer, when just starting to read this manuscript and it's already painful I don't want to read on – and I am a fan of these researchers.

Introduction

Line 19: delete "astonishing" – it's an opinion; opinions should be avoided

Line 21: change "allows to inject" to "allows for the injection of"

Line 22: change "also called" to "termed the"

Line 27: change "T3SS" to "injectisome"

Line 29: change "a series of ring structures" to "the injectisome basal structure"

Line 30: "incorporates", "a multimeric protein complex", delete "actual"

Line 32: delete "first" – also, it good to avoid using the same word more than once in a sentence; I suggest changing either "exports" to "secretes" or "export substrates" to "secretion substrates"

Line 38: insert "virulence-associated" before "type III secretion"

Line 39: delete "first"

Line 42: ?? – what happens to what upon the initiation of protein secretion??

Line 44: I am bothered by the term "sorting platform" – does it really sort – ie. bind all substrates equally and then sort (like the post office with letters) or does it have differential affinities for different substrates (the latter seems more plausible) in which case it should be called a "docking platform" or "affinity platform".

Line 47: there are 4 FliN molecules associated with one FliM (it's a tetramer)

Lines 47 – 50: This is a run-on sentence. I suggest breaking it down into multiple sentences. Where ever possible try to write distinct subject-verb-noun sentences. This will make your writing much easier to read and understand.

Lines 50 – 53: Another run-on

Lines 57-60: doesn't this sentence argue for a "affinity platform" and not a "sorting platform"?

Line 62: Based on Figure 1 it appears SctN is the base of the "sorting platform", and if they are not stably bound why are they called a platform?

Line 74: delete "so far"

Line 75: change "this" to "the" and "is" to "was"

Line 78: move "hereafter" before "called"

Line 81 delete "therefore"

Line 87: insert "us" before "to quantify"

Line 88: "bacteria"

Line 89" change "additionally" to "also"

Results

Chapter 1 binding of SctQ & L to effectors using PALM

Line 125: "The used construct" – painful to read; are you talking about multiple constructs here?; this needs better definition as the subject of this sentence.

Line 129: Why is the observation that PAmCherry displaying the fastest diffusion the expected

result??? You need to write for a general audience.

Line 130: define mean jump distance (mjd) in a sentence following this sentence and again write for a general audience that has never heard of mjd before (ie. how is it a measure of diffusion? And what does that mean?).

Line 134: delete "already"

Figure 1: you talk about a "sorting platform" yet you don't label it in your figure

Chapter 2: Effect of secretion activation and presence of effectors on mobility of sorting platform complexes

Line 155: A one-sentence paragraph, which is simply another run-on sentence.

I have decided that if I continue to work on the grammar I will never finish the review of the science – which is great. This manuscript needs serious editing before it can go further.

Author Rebuttal, first revision:

We thank all reviewers for their contributions and suggestions.

In response to the requests, we have performed a large number of additional experiments, including additional *in vitro* and *in vivo* interaction studies, fluorescence microscopy and single particle tracking, protein stability assays, and reanalyzed some of the existing data. For your information, we have summarized the new experiments, figures and tables below. We hope that the reviewers will agree that the additional experiments and other changes in this major revision, based on the reviewer comments, both further support the main findings of the manuscript and make them more accessible to the scientific community.

All individual responses to the reviewer comments are listed below in the detailed individual responses.

Summary of new and revised figures, tables and videos

	# initial	# revised	notes
Figures	1	1	slight changes in figure and legend
	2	2	slight changes in figure and legend
	3	3	
	4	4	
Tables		1	new – proximity labeling
	1	2	supplemented with apparent diffusion coefficients

Suppl. Figures	1	1	modified
	2	2	supplemented with additional protein stability data
	3	3	changes in legend
		4	new – in vitro interaction studies
	4	5	
	5	6	
		7	new – protein turnover
		8	new – quantification of cytosolic clusters
	6	9	
		10	new – cellular localization of effector and chaperone
		11	new – quantification of cytosolic fraction of SctQ
		12	new – determination of relative SctQ:SctL stoichiometry
	7	13	
		14	new – proximity labeling controls
Suppl. Tables	1	new – proximity labeling	
1	2	updated	
2	3		
3	4	updated	
	5	new – assignment of proteomics raw data in repository	
Suppl. Videos	1	new – raw sptPALM data	
	2	new – raw sptPALM data	
	3	new – raw sptPALM data	

Reviewer #1 (Remarks to the Author):

In this paper, Diepold and collaborators examined mechanisms by which type III secreted client proteins may be recruited to the secretion machine. Type III secretion systems (T3SSs) are composed of a bacterial- envelope-embedded structure known as the "basal body", and a cytoplasmic complex known as the C-ring (also known as the "sorting platform"). Cryo-ET studies have shown that the C-ring or sorting platform is composed of 6 pods, each one made of the SctK, SctL, and SctQ proteins. This structure provides a scaffold for a hexameric ATPase (SctN) to place it in close register to an enclosed nonameric ring structure composed of a single protein, SctV, which acts as the "export gate" and receptor for client proteins. The mechanisms by which translocases and effector proteins are recruited to the secretion machine are incompletely understood. The current model supported by a large amount of data states that effector-chaperone complexes reach the sorting platform by diffusion, bind to "receptor" proteins (most likely SctV, SctN, or both), and are initiated into the secretion pathway. Different affinities of the different client proteins for the receptors dictates the order in which they are secreted. This model accommodates the available data and the rather rapid secretion rate that has been measured in several systems. Diepold and collaborators, on the other hand, have proposed an alternative model in which the entire sorting platform assembles and disassembles during secretion, a process that, as expanded in this study, results in the recruitment of client proteins to the secretion machine after their shuttling by the components of sorting platform themselves.

This model was originally advanced based on diffraction-limited FRAP studies of a component of the sorting platform (SctQ) of the T3SS of the bacterial pathogen *Yersinia* but has not been corroborated for any other T3SS or by any other laboratory. The T3SS community has been sceptic about this model because of the "soft" nature of the original data (diffraction-limited FRAP), and its rather counter intuitive nature (i. e. how the cycle of assembly/disassembly of the C-ring could possibly keep up with the fast rate of secretion).

Furthermore, visualization of the T3SS structure by cryo ET has shown the presence of the sorting platform in almost 100% of the secretion machines, an observation that is in principle incompatible with the notion of continuous assembly/disassembly during its activity. These observations provide a context for the study described in this paper, where it is proposed that components of the C-ring (SctQ/SctL) shuttle effector proteins to the secretion machine. This conclusion is reached with remarkably little data to support it (see below). In addition, there are serious experimental shortcomings that raises question about the reliability of the imaging data obtained. Given the rather contested nature of the original claim, premature publication of these results without the appropriate evidence to support it would only add to the skepticism and would not serve well the authors or the T3SS research community at large.

We thank the reviewer for the detailed statement and comments. While not the focus of this manuscript, we feel that it is important to comment on some of the more general aspects the reviewer touches on.

It has been known for a long time that SctQ has a substantial cytosolic fraction in all tested bacteria, including *Shigella flexneri* (Johnson and Blocker, FEMS Microbiol Lett 2008), *Salmonella* SPI-1 (Lara-Tejero *et al*, Science 2011; Zhang *et al*, PNAS 2017) and *Yersinia enterocolitica* (Diepold *et al*,

Nat Commun 2017). This was observed by a diverse set of methods, including biochemical studies, fractionation, and classical fluorescence microscopy. Single particle tracking (Diepold *et al*, PLOS Biol 2015; Rocha *et al*, Integr Biol 2018; Prindle *et al*, Microbiol Spectr 2022) further described the behavior of these cytosolic complexes. The exchange of these two well-established populations was described by fluorescence recovery after photobleaching of natively expressed proteins, which is the gold standard in the field (see Tusk *et al*, JMB 2018, for a review). The results clearly show exchange of SctQ (but not of other T3SS components, SctC and SctV; Diepold *et al*, PLOS Biol 2015). While we appreciate that the more dynamic nature of protein complexes revealed by fluorescence microscopy and single particle tracking methods is challenging a more static view shaped by the classical structural biology that has advanced and shaped the field for many years, the role of protein dynamics in the T3SS has been appreciated by large parts of the community, as

documented by general T3SS reviews (e.g. by the Strynadka and Finlay groups (Deng *et al*, Nat Rev Microbiol 2017) and the Wagner group (Wagner *et al*, FEMS Microbiol Lett 2018)) and reviews focusing on other bacteria such as EPEC (Soto *et al*, J Bact 2017) and *Shigella flexneri* (Picking lab, Tachiyama *et al*, Front Cell Infect Microbiol 2021 and Muthuramalingam *et al*, Microorgan 2021). Indeed, exchange of SctQ at the injectisome can be exploited to gain light-dependent control over T3SS secretion, as has been recently published (Lindner *et al*, Nat Commun 2020) and patented by our research group.

The reviewer correctly mentions that in the predominant subtomogram averages of T3SS in many cryo-ET studies, the occupancy of the binding sites for the sorting platform is almost complete (although other reports, e.g. Nans *et al.*, Nat Commun 2015 and Makino *et al*, Sci Rep 2016 show fractions of unoccupied injectisomes or lower electron density in averages). This high occupancy is in line with our fluorescence microscopy data, which shows a high occupancy for all sorting platform components in live bacteria (Diepold *et al*, Nat Commun 2017). As already described in this earlier study, a high occupancy in no way means that there is no exchange; in other words, a binding site can be occupied most of the time and still show high exchange rates. Notably, the sorting platform was never copurified with the remaining injectisome “needle complexes” in any organism, which is highly compatible with a more dynamic, less static association.

We acknowledge that this background information, although not the focus of this study, should be provided in a concise form in order to make the manuscript more accessible to a broader audience (as also requested by reviewer 3). We have therefore recapitulated the existing data on protein exchange in different type III secretion systems in the introduction and discussion of the revised version of the manuscript.

Specific comments:

1) The entire premise that SctQ presumably working in conjunction with SctL act as a shuttle for the effector/chaperone complexes recruiting them to the C ring or sorting platform is supported by observations that the mobility of SctQ and SctL, as measured by single particle tracking PALM (sptPALM) is "slowed down" in the presence of effector proteins. It is assumed that such reduction in the diffusion of SctQ and SctL is due to being bound to the effector/chaperone complex. However, no direct evidence is provided to support this premise. This is a major shortcoming as, even if their measurements were to be correct (although see below), there are alternative explanations for their observations that cannot be ruled out with the data presented. In fact, the authors attempted to demonstrate direct-binding but they failed. They ascribe their failure to a proposed “transient binding” nature of the interaction. However, they only attempted to measure interactions by co-immunoprecipitation in the presence of detergents, which is a rather crude method to attempt to measure protein-protein interactions. There are multitude of experimental approaches that the investigators could have deployed to detect even the most transient of interactions. In the context of the failure of the T3SS community to detect interactions between C-ring components and effector proteins with even more sensitive approaches, without this crucial piece of evidence, the authors cannot draw the conclusions that they are drawing.

As noted in the manuscript and also by reviewer 2, interactions between the T3SS sorting platform components and effectors and their chaperones have been shown by various researchers with different methods and in different organisms, such as Morita-Ishihara *et al*. (JBC 2006), Spaeth *et al*. (PLOS Pathog 2009), and Lara-Tejero *et al*. (Science 2011). However, we agree with the reviewer

that other researchers, including ourselves in the present study, did not consistently detect these interactions by biochemical approaches. We attribute these method- and organism-dependent deviations to the transient binding inherent to transport processes. We agree that a better description of these interactions by independent experiments will support our conclusions and strengthen the manuscript, and therefore performed additional *in vitro* and *in situ* experiments. In line with our previous findings, biolayer interferometry, as an less sensitive method, did not allow to detect the interaction between the effector YopO (purified in presence of its chaperone SycO) and the sorting platform protein SctQ, whereas crosslinking-purification specifically detected their interaction, but not of either of these proteins to EGFP in a control experiment

(new **Suppl. Fig. 4**). To additionally test the interaction with another *in vivo* method, we used proximity labeling, a rather novel method that is especially suitable for transient interactions (Qin *et al*, Nature Methods 2021), doi 10.1038/s41592-020-01010-5). Our results, presented in the new **Table 1** and **Suppl. Table 1**, confirm the interaction of SctQ with its known interaction partner SctL, but also with several T3SS effectors, with high specificity.

2) An essential control missing in this study is the use of a YscQ mutant that is unable to bind effectors but retains the ability to bind sorting platform partners. This reviewer understands that without knowing whether YscQ even binds to client proteins to begin with, identifying a mutant may be challenging but to advance the hypothesis the authors are advancing with the experimental tools they are using, such a control is essential.

We agree that a distinct mutant that retains one binding interface while losing the other would be desirable. However, no such mutant has been described and it is unclear if such a mutant even exists. Perhaps more importantly, we already show the distinct effect of the presence of effectors on SctQ mobility, both in absence and in presence of the other sorting platform proteins (Fig. 1 and 2 of the manuscript, respectively). These experiments, including the relevant controls, show the interaction between effectors and SctQ in a clear and less invasive way. As indicated in the response to the previous point, we have now additionally tested the interaction using several technically unrelated approaches.

3) Surprisingly, no evidence is presented that indeed the EscQ/EscL complex co-localize with effectors in the cytoplasm as their model would predict; this is a crucial piece of data also missing from this study.

If we understand the request correctly, the reviewer questions the (co-)localization of the sorting platform components and the effectors in the bacterial cytosol. It has been known for a long time that effectors localize in the cytosol (see e.g. Jacobi *et al*, Mol Microbiol 1998; Enninga *et al*, Nat Meth 2005; Zhang *et al*, PNAS 2017 to list a few examples in different bacteria). Likewise, it has been shown repeatedly that the majority of SctQ and SctL proteins are soluble in the cytosol (Johnson and Blocker, FEMS Microbiol Lett 2008; Lara-Tejero *et al*, Science 2011; Diepold *et al*, Nat Commun 2017) and that they are completely cytosolic in the absence of basal body components (Diepold *et al*, EMBO J 2010; Diepold *et al*, Nat Commun 2017; Rocha *et al*, Integr Biol 2018). We also show the localization of SctQ in presence and absence of basal body components in this manuscript (**Fig. 2b**, **Fig. 3**). Nevertheless, as requested by the reviewer, we also performed an additional localization experiment of an effector and a chaperone in *Yersinia enterocolitica*. As expected, both proteins are also cytosolic. Although such a colocalization in the cytosol is not a specific indication for interaction, we now include our data as new **Suppl. Fig. 10**. For direct interaction experiments, we refer to our response to the previous points.

4) A unique aspect of the Yersinia T3SS is that its expression, assembly, and activation can be induced by specific growth conditions (used in this study; 37 °C in the absence of Ca). It should be stated that these conditions, while experimentally useful, are highly artificial so much so that it leads to bacterial growth arrest, a phenotype originally used by Brubaker, Straley, Goguen and others to identify the Yersinia T3SS genes in this bacterium (the so called Lcr or “low Ca response” genes). Since expression, assembly, and secretion are going on at the same time during the experimental conditions used in these studies, the authors cannot rule out that a proportion (or the totality) of the “movement” of SctQ/SctL detected by sptPALM may be

related to the journey from their place of synthesis in the cytoplasm to their destination to form the C ring. Even if the measurements were to be correct and SctQ's mobility in the cytoplasm is reduced in the presence of effectors, the authors cannot rule out an artifact stemming from YscQ being bound prematurely in the cytoplasm by the effectors prior to its recruitment to the C ring because the conditions used in the experiment are such that multiple events are going on at the same time: T3SS expression, injectisome assembly, and client protein secretion. Experiments must be done to uncouple these events. In other words, experiments should be done in which the injectisome is assembled first, and then effectors are expressed without de novo synthesis of YscQ. Otherwise, the multiple events cannot be untangled. The rather artificial

nature of the Yersinia system has led in the past to many artifactual observations that when observed in other systems, have not held up...this may well be another case..

While we understand that most researchers prefer certain (usually their own) model systems, we feel that some clarifications about the *Yersinia* T3SS are in place: Expression and assembly of the *Yersinia* T3SS are induced by a temperature shift to 37°C (mimicking entry into the host organism). Expression and assembly are therefore concurrent, like for the T3SS of other bacteria (and indeed almost any protein complex and molecular machine). Activation is then initiated by a different cue, host cell contact or calcium chelation, which is usually used in experiments. While it is unknown if and where the activation pathways for host cell contact and calcium chelation converge, the same is true for activation conditions used in the lab for other bacteria, such as Congo Red binding for *Shigella flexneri*. Likewise, the cessation of growth upon T3SS activation is not unique to *Yersinia*, but also present in two other main model systems, *Salmonella* SPI-1 (Sturm *et al*, PLOS Pathog 2011) and *Shigella flexneri* (Carter *et al*, Infect Immun 1980; Sasakawa *et al*, Infect Immun 1986). Activation of the system by host cell contact during infection will take place at 37°C, in a situation where assembly and secretion are active at the same time. Our experiments therefore more closely follow the relevant conditions and we feel that an artificial separation of assembly and a later activation of secretion under non-native conditions would not be helpful.

As an important side note, our data for interaction of SctQ and effectors covers (and is consistent in) both secreting and non-secreting conditions, as well as deletion strains, where there is no assembly of injectisomes at all (**Fig. 3**).

To refer to the second point, a possible role of freshly assembling SctQ molecules in an otherwise static structure is a scenario that can be easily addressed. Both our data and other researchers (e.g. Lara-Tejero *et al*, Science 2011) show that the majority of natively expressed SctQ is soluble in the cytosol. Our data shows a 15/85% distribution between injectisome-bound and cytosolic SctQ in the wild type strain (new **Suppl. Fig. 11**). While the number of injectisomes per cell does increase over time at 37°C (Kudryashev *et al*, Mol Microbiol 2015), this increase does not nearly match the measured exchange rate of SctQ. This rate (more precisely, the half-time of recovery in fluorescence photobleaching experiments (Diepold *et al*, PLOS Biol 2015)) is 68.2 ± 7.9 s. Based on the ratio of unbound SctQ (see above), at least one sixth of the cellular SctQ would be degraded and reproduced in this time, equivalent to a protein half-life of less than five minutes. While such a high turnover appears unlikely, we have quantified the protein stability of SctQ and other T3SS components after inhibition of new protein biosynthesis by addition of Tetracycline, an inhibitor of protein synthesis, under the conditions used in the experiments in this manuscript, and include the data as new **Suppl. Fig. 7**.

As noted above, the exchange of SctQ at the injectisome has additionally been confirmed independently by a recent application, in which light-dependent SctQ sequestration and release to and from the cytosol is used to control T3SS activity (Lindner *et al*, Nat Commun 2020). This patented method, which entirely relies on SctQ exchange, works very efficiently. The kinetics of light-dependent T3SS activation and deactivation are fully compatible with the exchange rates measured earlier (Diepold *et al*, PLOS Biol 2015).

5) In the absence of basal body components, it has been shown that EstQ aggregates, which can of course affect its mobility. This possibility has not been ruled out in these studies. Also, overexpression of SctQ can

lead to a proportion of molecules misfolded, which would also alter their mobility.

SctQ is not overexpressed in our experiments, but expressed from the native promotor, as is the case for all fusion proteins used in the manuscript with exceptions clearly noted (see main text and Material and Methods). While SctQ indeed shows aggregation (or at least polar localization) in absence of basal body components in *Salmonella* SPI-1 (Zhang *et al*, PNAS 2017), it does not aggregate in *Y. enterocolitica* (Diepold *et al*, Nat Commun 2017), as also visible in **Fig. 3**. This is further documented by the diffusion data itself: Aggregated proteins, e.g. in insoluble intracellular complexes and inclusion bodies, do not diffuse in the free,

Brownian manner observed (for comparison, see a study on Cas proteins, which do indeed aggregate into inclusion bodies for overexpression conditions, Fig. 6C in Turkowyd *et al*, Meth Enzymol 2019, <https://doi.org/10.1016/bs.mie.2018.11.001>). Notably, the observed diffusion data for cytosolic SctQ is highly similar in the wild-type and $\Delta sctD$ strain (**Fig. 2, 3**). Aggregated proteins would further be expected to no longer interact normally with other cytosolic components (which they do both in presence and absence of SctD, see Diepold *et al*, Nat Commun 2017) and no longer bind effectors (which they do, see **Fig. 3**).

In the one condition where we do not express the fusion proteins from their native genetic background, but from plasmid – in **Fig. 1** when testing the interactions of platform proteins with the effectors in the absence of the other T3SS components – we do not observe any aggregation artifacts. As stated in the manuscript, this is further supported by the fact that PAMCherry-SctQ mobility is very similar in these conditions (**Fig. 1b**) and when expressed in the native genetic background in absence of SctL (**Fig. 3**).

6) The data does not address the need for a shuttling protein that will “transport” effectors to the sorting platform when diffusion alone would just do the same more efficiently!! Effectors could then be simply “trap” by the sorting platform. In other words, diffusion will always outpace even the fastest secretion machine. The model proposed is not only counter-intuitive but inconsistent with the rapid speed of secretion.

We thank the reviewer for raising the point of the potential biological benefit of the proposed mechanism. While the focus of this manuscript is the description of the observed molecular phenotype, we have thought about possible – and so far absolutely speculative – benefits of the binding of effectors to the sorting platform proteins. As mentioned by the reviewer, the small size of bacteria allows effector proteins (just like any other freely diffusing protein in the cell) to reach any other point in the cell within sufficiently short time to explain the reported export rates. Although several researchers have reported interactions of effectors with sorting platform components in different organisms (Morita-Ishihara *et al*, JBC 2006; Spaeth *et al*, PLOS Pathog 2009; Lara-Tejero *et al*, Science 2011), and similar two-step binding processes are known for sec- based protein export and the export of the T4SS relaxase, their functional relevance for the T3SS has remained unclear. One possible explanation, however, is an increase in export specificity. T3SS export displays an extraordinarily high fidelity (>90% of all proteins in the supernatant of secreting *Yersinia* are *bona fide* T3SS export substrates), despite a lack of specific motifs in the secretion signal sequences targeting effectors (and other proteins, when fused) to the injectisome. A key question is how such a high specificity is reached despite the expected small differences in binding energy between T3SS effectors and non-exported proteins with closely related sequences. A series of subsequent binding events in different cellular context – e.g. cytosolic effector binding to SctQ/L, followed by highly specific binding to SctV at the injectisome – could be a basis of this specificity. Alternatively, a shuttling model could provide access for SctQ/L-bound proteins to the restricted environment of the cage-like structure at the proximal side of the export gate, bounded by SctQLN. While additional experiments are clearly required to test such models, they are in line with both the binding of effectors to the sorting platform and with the high substrate specificity observed for the T3SS. We now discuss this and other possible biological benefits of the described molecular phenotype – clearly stating their speculative nature – in the revised manuscript.

7) The mobility measurements have been done in 2D. However, diffusion occurs in 3D. Consequently, the actual rates measured do not reflect reality. Measurements should be done in 3D. It is unclear why the authors have chosen to do them in 2D.

Brownian diffusion is independent in all three dimensions, i.e. the motion of the diffusive particle in one dimension is independent of its direction into another dimension (Einstein, Ann Phys 1905). So, for measuring Brownian diffusion it does not matter if one measures 1D, 2D, or 3D. Only directed motion or diffusion in a gradient is not isotropic and thus may need to be measured in 3D. A more relevant aspect is the limited space in bacteria, which impacts 2D and 3D measurements alike. For this reason, we explicitly state jump distances and not diffusion coefficients in the manuscript (see also response to Reviewer 2,

question 3). Importantly, existing methods for 3D imaging in PALM come with disadvantages, especially in terms of limited and highly anisotropic resolution. For this work, we therefore decided actively to make use of the higher localization precision of fluorescent spots in 2D imaging.

8) The methodology is very poorly described and with the information given, it would be impossible, even for a laboratory with a lot of experience in high-end imaging to reproduce these studies. For example, for some experiments the authors state that they have eliminated information related to molecules present at the membrane. Yet, the way the experiments were conducted, it would be impossible to identify pixels at the membrane since bacteria are 3 D objects and the way the images appear to have been collected, it would be impossible to distinguish molecules in the cytoplasm from those at the membrane on top or below. It should be said that sptPALM was originally described by Lippincott-Schwartz and Betzig to investigate movement of molecules at the membrane, which is a 2D environment and not a 3D environment like the one is being imaged in these studies. Also, it is unclear whether only 1 jump was measured or multiple jumps. In that case, new molecules being photo-activated would interfere with the interpretation of the results.

We thank the reviewer for this comment and have rewritten the methodology part of the Material and Methods section. The revised version includes the information requested by the reviewer and we hope that it is more easily understandable, both for experts and non-experts. We also include exemplary video raw data as new **Suppl. Videos 1-3**, so that the data analysis can be completely reassessed by any reader.

9) The conclusions drawn related to the composition/stoichiometry of a single pod are highly speculative are not supported by the data. No conclusions can be drawn as the approach on stoichiometry of complexes with the approach used in these studies. Inferences made from the values stated in **Table 1** cannot possibly be correct since the authors have no information about the real composition or shape of the complexes. Yet, an inexperienced reader may get the idea that those values reflect some reality, which it is not the case. The authors do not help by making aggressive claims not supported by the data.

As stated by the reviewer, the composition of the sorting platform is currently unclear – researchers have come to deviating conclusions in different organisms using different methods. Fluorescence-based studies by our lab in *Yersinia* (Diepold *et al*, PLOS Biol 2015 and Nat Commun 2017) and by the Galán lab in *Salmonella* SPI-1 (Zhang *et al*, PNAS 2017) indicated ~24 copies of SctQ per injectisome. In contrast, the main peaks detected in a native mass spectrometry / small angle scattering study in *Salmonella* SPI-1 indicated 1 or 2 SctQ per detected substructure, corresponding to 6 or 12 SctQ copies per injectisome (Bernal *et al*, JMB 2019). Another recent study by the Galán group based on photocrosslinking (Soto *et al*, PNAS 2022) detected one SctQ per SctK for *Salmonella* SPI-1, which would again correspond to 6 SctQ per injectisome – although the authors explicitly state the possibility of additional SctQ molecules bound in different conformation and/or dynamically. Indeed, such variances in results from different methods would not be surprising for transient complexes, where *in vivo* methods and methods based on the detection of more stable complexes could produce different results (e.g. in case of transient association of molecules with a stable core, as suggested in Soto *et al*, PNAS 2022).

We now more clearly point out these different studies in the introduction and the discussion of the revised version. In addition, we rephrased the legend to Table 1 (now **Table 2**) and also include alternative proposed stoichiometries to allow the reader to appreciate the different models and to

directly evaluate the compatibility of our data with these models.

Importantly, we have performed additional experiments using photoactivatable localization microscopy to directly count SctQ and SctL molecules, both in live and in fixed bacteria. The results, which are included as new **Suppl. Fig. 12**, clearly indicate a 2:1 stoichiometry of SctQ:SctL in live *Y. enterocolitica*. Given the presence of 12 copies of SctL per injectisome (which all current models agree on), this strongly supports a copy number of 24 for SctQ, in line with previous results for SPI-1 (Zhang *et al*, PNAS 2017) and our lab for *Y. enterocolitica* (Diepold *et al*, PLOS Biol 2015 and Nat Commun 2017).

7) The Δ effector strain used in some of the studies still retain the protein translocases. Yet, the authors observed an apparent difference in the mobility of SctQ in comparison with wild type, a difference that is equivalent to what they observed in the absence of the plasmid (i. e. in the absence of translocases and effectors). This makes little sense even in the context of the model presented by the authors since the translocases would also bind SctQ and slow it down. Secretion of the protein translocases is equally dependent on the C ring components and there are no data indicating that their engagement by the secretion machine occurs by different mechanisms. These observations cast serious doubts about what the authors are actually measuring or what does it all mean...

The reviewer is right about the presence of the translocators in the Δ effectors strain; we mention this at the beginning of the respective paragraph of the results section and have even quantified the proportion of translocators in **Suppl. Table 2**. As shown there, the hydrophilic translocator, SctA, accounts for 6-9% of all exported proteins, depending on the conditions, while the pore-forming hydrophobic translocators (which cannot be deleted without deregulating the system, see Williams and Straley, J Bact 1998, Wang *et al*, Science 2016) make up 16-34%. All this is in line with our observation that even in the Δ effectors strain, peaks corresponding to larger complexes are still visible, albeit to a lower degree (**Fig. 2d**). We now specifically mention this point as a footnote and hope that this helps readers interested in this particular binding event to appreciate the data.

8) If binding of SctQ to the effectors is so transient, how do the authors consider into their trajectories the kinetics of unbinding?

The SPT-PALM measurements are highly time-resolved (with the majority of molecules being tracked for

<200 ms). Within this short time, we do not expect a significant proportion of binding and unbinding events to occur and indeed, we do not see this change of molecular state in our data. A change of state would be visible by an increase (unbinding) or decrease (binding) of the jump distances. Our tracking tool *swift* is programmed to detect statistical changes of molecule behavior, e.g. changes in diffusion characteristics (slower/faster), changes in brightness (particle merging, e.g. dimerization). It in such a case *swift* splits the trajectory into two so called segments (as an intermediate level in between localizations and trajectories, so e.g. 10 localizations in 2 segments in 1 trajectory). Our cytosolic data (with N in the several 10,000s) does not contain trajectories with two segments, while the injectisome immobile data set contains a handful (for several hundreds of immobile trajectories in the WT data). This is expected for short trajectories (see Persson *et al*, Nat Meth 2013 for a detailed discussion). As mentioned earlier, transient binding is not at odds with the high proportion of binding (in this case the high proportion of SctQ bound to effectors that we observe).

9) How do the authors explain the much slower mobility of SctL (25kDa) compared to the bigger proteins SctQ (37 kDa) and SctN (47 kDa)? Even if one takes a look at Table 1, the mobility peak of PAmCherry-SctL with no effectors (~300 nm) is comparable to much bigger "predicted" complexes. Makes little sense...

We assume that the reviewer refers to **Figure 1**, the analysis of the labeled proteins in the strain cured of the virulence plasmid and therefore lacking other T3SS components. Based on the available data for the multimerization state of the tested proteins, the expected molecular weight of the predominant diffusing units without bound effectors would be 107.2 kDa for SctL (PAmCherry-SctL₂), 82.5 kDa for SctQ

(PAmCherry-SctQ-SctQ_{C,2} complex; Bzymek *et al*, Biochem 2012; McDowell *et al*, Mol Microbiol 2016), and

77.3 kDa for SctN (PAmCherry-SctN). The increasing mobility observed for SctL<SctQ<SctN (**Fig. 1bc**) therefore does match the expected order. While the molecular weights of these units has already been listed in the original submission, we now mention these facts more clearly in the results and discussion sections.

Reviewer #2 (Remarks to the Author):

Type III secretion systems are widespread in bacteria and they are used to inject a variety of effector proteins into eukaryotic cells. The structure and composition of T3SS is conserved and also the flagella apparatus shares the conserved structure and functions. In recent years a wealth of new data on the structure and assembly of T3SS has been published. However, the assembly dynamics, as in most cases of large cellular complexes, remained insufficient understood. In particular interaction of the secreted substrates with the cytosolic sorting platform is unclear and under discussion. The authors use here single particle tracking (sptPALM) to unravel dynamics of sorting platform proteins and substrates in presence and absence of components in live *Yersinia enterocolitica* cells. They complement their analyses with co-immunoprecipitation and mass spectrometry data. Overall, this work is a nice contribution to the field and the experiments are carefully executed and written up in a concise and coherent manuscript. In particular, the SPT data are very impressive and of high quality. A weak point is that the major claims are based on assumptions and calculations of theoretical diffusion of proteins and complexes. While I understand that there are structural data and other biochemical data that suggest existence of certain sorting platform sub complexes, the data are rather indirect. Below I have summarized a couple of points that might be helpful to revise the manuscript.

We thank the reviewer for the thorough and positive review! Our replies and additional experiments in response to the specific points raised are listed below.

Main points:

Supplementary Figures 1, 2 and 4 show immunoblot data of PAmCherry labelled T3SS proteins. I am puzzled by the differences in stability. PA-mCherry-SctQ in Fig. S1 appears reasonable stable as a full length band, migrating above the 72 kDa marker. In Fig. S2a the same construct appears well below the 72 kDa marker (red asterisk). In line 1 there is barely any full length protein visible, while a clear degradation band is prominent. The elution fraction shows two major bands. Although there seems no free PA-mCherry, the existence of degradation products will complicate the SPT analysis. Quite similar is the problem for SctL and SctN (Fig. S2B-C).

Thanks for raising these points, which are indeed important for the analysis.

During the time we were conducting the experiments for this manuscript, we changed our standard protein standard (from BioRad All Blue Prestained to Jena Biosciences Classic Blue), which has led to initial mis-annotations in some cases. We are very sorry for this lapse, highly thankful for the reviewer pointing at this, and have gone through all the raw data from all Western Blots, checked the annotation and repeated the experiment, where necessary. In **Suppl. Fig. 1**, the annotation was off by one band, for which we apologize. We have now corrected the annotation. **Suppl. Fig. 4** (now **Suppl. Fig. 5**) shows stability of the proteins in the absence of their native interaction partners, in the strain lacking the virulence plasmid. It is possible that the proteins are more prone to limited degradation in this case. However, a series of additional experiments performed for the revised version have consistently shown less degradation of the proteins in the strain lacking the virulence plasmid, and we have therefore included a more representative immunoblot in **Suppl. Fig. 4** (now

Suppl. Fig. 5). We suspect that the degradation visible in **Suppl. Fig. 2**, especially for PAmCherry-SctQ, occurs during the time-consuming handling of the pellets from the large used culture volumes prior to and during cell lysis, despite the presence of protease inhibitor. To test this, we ran an additional experiment, where samples were taken at different time points of this preparation. The results support this hypothesis, and we now include this data as a **new panel d in Suppl. Fig. 2**. Although a comparison of the samples taken during purification (purification steps 1-4 in **Suppl. Fig. 2a-c**) indicates that this lysis does not proceed during the further purification, this adds to the fact that the co-IP/proteomics analysis is qualitative

rather than quantitative, as correctly mentioned by the reviewer (see response below). In all cases, we now indicate the expected molecular weight in the figures and legends.

Supplementary Fig. 3 shows the data of mass spectrometry analysis with SctQ, SctL and SctN as bait proteins. I did not find any hint of where the mass spec raw data were deposited in a database. While the interaction of the sorting platform proteins with substrate proteins (Yop) is clear (also from previous publications), I am not sure how these data are in agreement with the final model. There the Yop proteins are indicated to bind to SctQ only, while the mass spec data also suggest interaction with SctLN in some cases. Using Mass spec data to quantify interaction strength is in my opinion to much speculation. Protein-protein interactions can quantitatively be addressed by various methods including SPR, BLI, ITC etc., but that would require more in vitro work. I therefore suggest to discuss this better in the text and tone down the statements.

As requested, we have now deposited the raw mass spectrometry data in the PRIDE database (project accession PXD044214). We will release the data upon publication of the manuscript and have already made it accessible to the reviewers using the user name reviewer_pxd044214@ebi.ac.uk and the password *GhARVZIX*. For the interaction of effectors with the sorting platform proteins, we absolutely agree that the results are heterogeneous, which is why we chose to study these key interactions in live cells in the first place. We picked the “+” – “+++” classification for the same reason - to avoid a false impression of a binary “interaction / no interaction” result. The legend states that this represents the significance of enrichment, rather than a quantification of interaction strength. We now aim to make this point even clearer by consistently mentioning that this classification is based on the significance of enrichment in co-immunoprecipitation and does not quantify any strength of interaction. We also provide the link to the depository of the raw proteomics data at the end of the manuscript and provide an overview in the new **Suppl. Table 5**.

Importantly, we agree with the reviewer that additional experiments to test the interaction of effectors and chaperones with SctQ are the best solution to further strengthen and quantify this main point of our manuscript. To this aim, we purified SctQ and the effector YopO (in presence of its chaperone SycO) in

E. coli. As suggested by the reviewer, we used biolayer interferometry and chemical crosslinking to detect the interactions. While biolayer interferometry did not allow to detect the interaction between the effector YopO and the sorting platform protein SctQ, crosslinking-purification allowed to specifically detect the interaction of YopO and SctQ (but not of either of these proteins to EGFP, used as a control). These results, which are presented in the new **Suppl. Fig. 4**, are in line with the previous interaction results (**Fig. 1**) suggesting that the interactions are transient. In addition, we also used a second *in situ* interaction method, proximity labeling, which is especially suitable for transient interactions. This method very specifically shows the interaction of SctQ with the majority of *Yersinia* effectors (and its known interaction partner SctL) in live bacteria. The results are shown in the new **Table 1** and **Suppl. Table 1**.

The dynamic behavior of the T3SS proteins is estimated based on a jump distance analysis. This is a very valid procedure, but it does not allow comparison with other protein diffusion data. I assume the authors relied on Jump distance analysis because all other calculations of diffusion coefficients are error prone. Still I was wondering if there is a chance at all to compare diffusion data between different experiments to allow a comparison between the new and published data on other

complexes. In particular, since the authors used these coefficients in their calculations. This could maybe clearer in the text.

As correctly assumed by the reviewer, we chose not to rely on long-range analyses like mean squared displacement (MSD) calculations due to the confinement of proteins within the relatively small bacterial cells. However, we can extract a diffusion coefficient D from the mean jump distance analysis, which we have already done in a previous publication for comparative reasons (Wimmi *et al*, Nat Commun 2021). To estimate the diffusion coefficient D from the MJDs (M) at the frame rate τ (15 ms for this study) and the localization precision σ_m (median of 15.6 nm, average of 16.5 +/- 4.4 nm for this study), the equation

$D = 3/2(M^2\tau^{-1} - \sigma_m^2)\tau^{-1}$ can be used. We are happy to do this for comparative reasons, even though results

from other groups mostly use MSD analysis to obtain a diffusion coefficient, which will be biased by confinement for faster diffusion, e.g. already at the level of single pods. We now include this comparison in Table 2 and added a brief additional information (**Suppl. Text 4**).

I agree that transient interactions of proteins and complexes are difficult to capture in live cells. However, there are several biochemical and genetic ways to overcome these limitations. Cross-linking by introducing cysteine residues at critical sites has been extensively used to stabilize transient interaction. It also offers a nice confirmation of structural organizations. This is of course involving a lot of experimental effort. An easier, but less precise way is crosslinking with fixatives.

We agree with the reviewer that this is a critical point. As mentioned above, we have performed additional *in vitro* and *in situ* experiments, including chemical crosslinking, to test these transient interactions (new **Suppl. Fig. 4, Table 1, Suppl. Table 1**).

Would the high resolution of SMLM be sufficient to gain insight into the structure of the T3SS sorting platform and the interaction with the injectisome? Ideally, using dual color SMLM could bring by additional information on spatial arrangements.

This would be tempting and indeed, for more distant components of the system, such as the translocon on the tip of the needle and the cytosolic domain of SctD, which are >100 nm apart, this has already been done by structured illumination microscopy (Nauth et al, PLOS Pathog 2018). Unfortunately, for the structure of the sorting platform itself, where multiple copies of the involved proteins are participating in a small volume with a diameter of approximately 30 nm and a height of approximately 20 nm, we expect that the localization precision of PALM (16.5 +/- 4.4 nm, see above) makes it very difficult to draw any conclusions about the internal structure of the sorting platform. This is also demonstrated by the fact that even the even higher resolution of MINIFLUX only yielded comparably little insight into the structure of SctL at the injectisome (Carsten *et al*, Meth Appl Fluoresc 2023, doi 10.1088/2050-6120/aca880).

Minor points

Line 164: What do the authors mean with "almost" no effectors?

YopQ, an effector with no known virulence function that is thought to act as a regulator of translocation rate (Dewoody *et al*, Front Cell Inf Micro 2013) is still present in the Δ effectors strain. We now mention this specifically at this point in the manuscript and refer to **Suppl. Table 3**, where this is shown and quantified.

Line 271: remove "earlier"

Figure 3 legend: line 295 - the last band is labelled "green" not blue as in the text. Thanks for the notice. We have corrected both points.

Line 324: Therefore, we in situ analyzed – rephrase

With the addition of the proximity labeling data, this paragraph was completely rewritten.

Lines 388: This work is not the first use of SPT to study protein interaction and dynamics in vivo. This

may be toned down.

We did not intend to claim that this is the first SPT study that looks at protein interaction and dynamics *in vivo*. However, to our knowledge, this is the first (or at least one of the first) study that explicitly uses it for biochemistry in the sense that we can distinguish different sub-complex variants by their diffusion in the cytosol (and e.g. not the classic binding to a macromolecule like DNA, resulting in the clearly distinguishable categories diffusive vs. immobile). We wanted to emphasize that SPT now is at the level of technical

precision allowing to do in vivo biochemistry, e.g. it can be very useful for projects where individual proteins cannot be purified and thus, classic in vitro biochemistry is not possible. As suggested by the reviewer, we carefully checked the respective text passages and rephrased them accordingly.

Line 427: The authors speak about "rearrangements", this should be better explained.

We have rewritten this paragraph and simplified it to more directly focus on the most relevant findings and their interpretation.

Line 510: Quantification of foci: The text reads as if all stable foci that are not at the outline of the cell were not shown. How many of these clusters were observed? If they are there, what are they?

We thank the reviewer for pointing out the unclear wording at this point. To be clearer, we have now explicitly quantified the fraction of tracks we did not select, so all foci that are not at the outline of the cell but in the cytosol. We did this both for a wild-type strain where, due to 3D projection effects of injectosome, clusters might actually sit at the membrane but appear to be intracellular, and the $\Delta sctD$ strain, which we already used as a control for the quantification of foci (to control for unspecific fluorescence). As can be seen in the new **Suppl. Fig. 8**, there are extremely few to no such foci, showing that neither noise nor unspecific binding of PAmCherry-SctQ affects our analysis. We specifically mention this points in the revised version.

Reviewer #3 (Remarks to the Author):

The significance of this work is that it takes us from conjecture that the sorting platform shuttles effector proteins to the injectisome to the actual data to support the hypothesis that this actually occurs (and quantitatively no less!). The authors combine various experimental approaches to provide important insights into the initiation of the secretion process and sequestration of effectors prior to their secretion and translocation into target host cells. The first chapter describes interactions between PAmCherry-SctQ PAmCherry-SctL and PAmCherry-SctN and the YopO/SycO effectors. The data is convincing for interactions with SctQ and SctL and not SctN, but the authors do not describe the methodology very well for a non- expert. Chapter 2 analyzes the mobility of sorting platform under secreting and on-secreting conditions using the PAmCherry-SctQ fusion and YopO/SycO substrates by the same method. The data is solid and clearly show ratio of effector/SctQ increases significantly compared to effector-free condition from non to secreting bacteria. Finally, they show binding of effectors to the sorting platform in live bacteria in the cytosol followed by convincing experiments that SctK&L, while apart of the sorting platform are not required to bind effectors, again in live individual cells.

This is actually a very cool and important study with very cutting-edge methodology employed. Please make the writing what it should be for such quality science: a pleasure to read.

I made minor comments through line 155, but stopped for lack of time. This needs editing, preferable by someone not in the field so that the targeted average reader of this Journal, a broad-audience journal, can understand the science, which first requires that the numerous grammatical problems be addressed.

We thank the reviewer for the very positive assessment of the science presented in the manuscript! In order to improve the grammar and overall readability of the manuscript, we have sought advice from several native English-speaking colleagues inside and outside the field and have implemented their changes, including all of the points specifically raised by the reviewer below.

Abstract

Line 2: change “The bacterial type III secretion system, also called the injectisome, translocates” to “Bacterial type III secretion is utilized by injectisome structures to translocate”.

The bacterial type III secretion system is not called the injectisome. Both injectisomes and flagella utilize type III secretion to secrete proteins.

We thank the reviewer for this comment and have changed the abstract accordingly; the nomenclature is now also defined in the first paragraph of the introduction.

Lines 7-8: this sentence makes no sense. I believe you are trying to say “There are ~5 injectisomes per cell under non-secreting growth conditions and ~18/cell under secreting growth conditions.”

Lines 8-9: another non-comprehensible sentence. Who are “they”? A mobile cytosolic pool of what? There are eleven authors on this manuscript, does anyone understand English grammar? You have

to understand that, as a reviewer, when just starting to read this manuscript and it's already painful I don't want to read on – and I am a fan of these researchers.

Additional experiments done for the revision (new **Suppl. Fig. 11**) now allow us to be more precise in the measurements – and hopefully in the expression of their results. Based on the suggestion of the reviewer, we changed this part of the abstract to “While 15% of the sorting platform proteins bind to injectisomes (~5 injectisomes per cell under non-secreting conditions, ~18 injectisomes per cell under secreting conditions), 85% were mobile in the cytosol.”

Introduction

Line 19: delete “astonishing” – it’s an opinion; opinions should be avoided

Changed.

Line 21: change “allows to inject” to “allows for the injection of”

Line 22: change “also called” to “termed the”

Both changed in the context of more clearly defining the T3SS and its role in the flagellum and injectisome at this place.

Line 27: change “T3SS” to “injectisome”

Line 29: change “a series of ring structures” to “the injectisome basal structure”
Line 30: “incorporates”, “a multimeric protein complex”, delete “actual”

Line 32: delete “first” – also, it good to avoid using the same word more than once in a sentence; I suggest changing either “exports” to “secretes” or “export substrates” to “secretion substrates”

Thanks for the suggestions; all changed.

Line 38: insert “virulence-associated” before “type III secretion”

Line 39: delete “first”

Both changed as suggested.

Line 42: ?? – what happens to what upon the initiation of protein secretion??

Changed to “do the interactions between effectors and T3SS components change upon the initiation of protein secretion, e.g. by host cell contact?”

Line 44: I am bothered by the term “sorting platform” – does it really sort – ie. bind all substrates equally and then sort (like the post office with letters) or does it have differential affinities for different substrates (the latter seems more plausible) in which case it should be called a “docking platform” or “affinity platform”.

We agree with the reviewer that the designation of the cytosolic T3SS components as a “sorting platform” is based on rather limited evidence and may be misleading. However, the name has become established in the field and we wanted to keep the manuscript accessible to all readers without establishing an additional name in addition to the concepts drawn out in the manuscript. Nevertheless, based on the suggestion of the reviewer, we now suggest to look at these components as a “docking complex” in the discussion.

Line 47: there are 4 FliN molecules associated with one FliM (it’s a tetramer)

Several reports using different organisms and techniques (including *in situ* structural data) point

towards a 1:3 stoichiometry of FliM:FliN (Notti *et al*, Nat Commun 2015; McDowell *et al*, Mol Microbiol 2016; Carroll *et al*, eLife 2020) and suggest that one FliM molecule, specifically its SPOA domain, replaces one FliN in the FliN tetramer found *in vitro*. However, other reports (Brown *et al*, 2005, Sarkar *et al*, 2010; Delalez *et al*, 2014) support a 1:4 stoichiometry. We now mention both possible stoichiometries and the references.

Lines 47 – 50: This is a run-on sentence. I suggest breaking it down into multiple sentences. Where ever possible try to write distinct subject-verb-noun sentences. This will make your writing much easier to read and understand.

Lines 50 – 53: Another run-on

We broke down both sentences into smaller parts. We tried to do the same throughout the manuscript. Lines 57-60: doesn't this sentence argue for a "affinity platform" and not a "sorting platform"?

Line 62: Based on Figure 1 it appears SctN is the base of the "sorting platform", and if they are not stably bound why are they called a platform?

As mentioned above, we agree, but will use the established term "sorting platform" in the results parts to keep it easy to read for the community. In the discussion, we now suggest to see the proteins as a mobile docking complex, in line with our results.

Line 74: delete "so far"

Line 75: change "this" to "the" and "is" to "was" Line 78: move "hereafter" before "called"

Line 81 delete "therefore"

Line 87: insert "us" before "to quantify"

All changed, thank you.

Line 88: "bacteria"

In agreement with native-speaking colleagues, we leave "per bacterium" in the singular form, but change "...per non-secreting bacterium..." to "...per bacterium under non-secreting conditions".

Line 89" change "additionally" to "also"

Changed.

Results

Chapter 1 binding of SctQ & L to effectors using PALM

Line 125: "The used construct" – painful to read; are you talking about multiple constructs here?; this needs better definition as the subject of this sentence.

Changed to "In bacteria with a functional T3SS, the plasmid used for the expression of YopO and SycO allowed for the secretion of YopO, indicating that YopO and SycO are expressed and functionally interact (Suppl. Fig. 4)."

Line 129: Why is the observation that PAmCherry displaying the fastest diffusion the expected result??? You need to write for a general audience.

We thank the reviewer for the justified comment. We now explain the reasoning at this point, but –

perhaps more importantly – also earlier, when we introduce the method.

Line 130: define mean jump distance (mjd) in a sentence following this sentence and again write for a general audience that has never heard of mjd before (ie. how is it a measure of diffusion? And what does that mean?).

We now define the mjd and explain its meaning in more detail when we introduce the method. Line 134: delete “already”

Deleted.

Figure 1: you talk about a “sorting platform” yet you don’t label it in your figure

We now label the sorting platform (as originally defined by Lara-Tejero and colleagues) in the figure. Chapter 2: Effect of secretion activation and presence of effectors on mobility of sorting platform complexes Line 155: A one-sentence paragraph, which is simply another run-on sentence.

As suggested, we broke down the sentence into two shorter sentences.

I have decided that if I continue to work on the grammar I will never finish the review of the science – which is great. This manuscript needs serious editing before it can go further.

We went through the entire manuscript with our native-speaking colleagues and made changes in several places. We placed particular emphasis on shortening and simplifying long sentences. We hope that this will make the manuscript much easier to read and allow readers to appreciate the results and their implications.

Decision Letter, second revision:

Message: Our ref: NMICROBIOL-22092364B

9th October 2023

Dear Andreas,

Thank you for your patience as we’ve prepared the guidelines for final submission of your Nature Microbiology manuscript, "Direct binding of type III secretion effectors to cytosolic sorting platform complexes in live bacteria indicates an effector shuttling mechanism" (NMICROBIOL-22092364B). Please carefully follow the step-by-step instructions provided in the attached file, and add a response in each row of the table to indicate the changes that you have made. Ensuring that each point is addressed will help to ensure that your revised manuscript can be swiftly handed over to our production team.

We would like to start working on your revised paper, with all of the requested files and forms, as soon as possible (preferably within one week). Please get in contact with us if you anticipate delays.

1When you upload your final materials, please include a point-by-point response to any remaining reviewer comments.

In recognition of the time and expertise our reviewers provide to Nature Microbiology's editorial process, we would like to formally acknowledge their contribution to the external peer review of your manuscript entitled "Direct binding of type III secretion effectors to cytosolic sorting platform complexes in live bacteria indicates an effector shuttling mechanism". For those reviewers who give their assent, we will be publishing their names alongside the published article.

Nature Microbiology offers a Transparent Peer Review option for new original research manuscripts submitted after December 1st, 2019. As part of this initiative, we encourage our authors to support increased transparency into the peer review process by agreeing to have the reviewer comments, author rebuttal letters, and editorial decision letters published as a Supplementary item. When you submit your final files please clearly state in your cover letter whether or not you would like to participate in this initiative. Please note that failure to state your preference will result in delays in accepting your manuscript for publication.

Cover suggestions

COVER ARTWORK: We welcome submissions of artwork for consideration for our cover. For more information, please see our [a href=https://www.nature.com/documents/Nature_covers_author_guide.pdf target="new">guide for cover artwork](https://www.nature.com/documents/Nature_covers_author_guide.pdf).

Nature Microbiology has now transitioned to a unified Rights Collection system which will allow our Author Services team to quickly and easily collect the rights and permissions required to publish your work. Approximately 10 days after your paper is formally accepted, you will receive an email in providing you with a link to complete the grant of rights. If your paper is eligible for Open Access, our Author Services team will also be in touch regarding any additional information that may be required to arrange payment for your article.

Please note that *Nature Microbiology* is a Transformative Journal (TJ). Authors may publish their research with us through the traditional subscription access route or make their paper immediately open access through payment of an article-processing charge (APC). Authors will not be required to make a final decision about access to their article

until it has been accepted. [Find out more about Transformational Journals](https://www.springernature.com/gp/open-research/transformational-journals)

Authors may need to take specific actions to achieve [compliance with funder and institutional open access mandates](https://www.springernature.com/gp/open-research/funding/policy-compliance-faqs). If your research is supported by a funder that requires immediate open access (e.g. according to [Plan S principles](https://www.springernature.com/gp/open-research/plan-s-compliance)) then you should select the gold OA route, and we will direct you to the compliant route where possible. For authors selecting the subscription publication route, the journal's standard licensing terms will need to be accepted, including [self-archiving policies](https://www.nature.com/nature-portfolio/editorial-policies/self-archiving-and-license-to-publish). Those licensing terms will supersede any other terms that the author or any third party may assert apply to any version of the manuscript.

For information regarding our different publishing models please see our [Transformational Journals](https://www.springernature.com/gp/open-research/transformational-journals) page. If you have any questions about costs, Open Access requirements, or our legal forms, please contact ASJournals@springernature.com.

Please use the following link for uploading these materials:
[redacted]

Best regards,
[redacted]

Reviewer #2:

Remarks to the Author:

The paper submitted by the Diephold and Endesfelder labs is a largely revised version. Their work shows the direct binding of Type III effector proteins to the sorting platform in the cytoplasm of *Yersinia enterocolitica*.

The thoroughly revised work contains several new experiments that mainly addressed the putative interaction of SctQ and effector proteins, protein stability and turnover as well as new data to solve the stoichiometry of the complexes.

Overall the authors have done a terrific job with this revision. I particularly like the organized answer with the table showing the revised figures - that made life easier during revision.

I am glad that the apparent protein stability problem in the first submission turned out to be just a labelling error. It is now shown that the fusion proteins are stable (enough) for the SMLM analysis.

The newly added proximity labelling experiments clearly strengthen the paper and confirm

3the interaction of SctL and SctQ. The proximity labelling does not reveal interaction of SctQ with YopD, which was tested in vitro by cross-linking. This needs explanation. Although, I appreciate the fact that the authors put a lot of effort in confirming interactions, the in vitro crosslinking is rather problematic. Unlike in vivo crosslinking, in vitro at sufficiently high protein concentrations often protein crosslinks can be achieved that are not found in vivo. Since this is just another control here, I do think this experiments can remain, but it needs to be taken with care and is not ultimate proof for interaction. I had problems opening the supplemental videos and I wonder whether there is a way to compress them so that they can be published as supplemental material alongside the paper, rather than being on an institutional webpage.

Reviewer #3:

Remarks to the Author:

My criticisms have all been adequately addressed, thank you!

Author Rebuttal, second revision:

Comments Reviewer 2

The paper submitted by the Diephold and Endesfelder labs is a largely revised version. Their work shows the direct binding of Type III effector proteins to the sorting platform in the cytoplasm of *Yersinia enterocolitica*.

The thoroughly revised work contains several new experiments that mainly addressed the putative interaction of SctQ and effector proteins, protein stability and turnover as well as new data to solve the stoichiometry of the complexes.

Overall the authors have done a terrific job with this revision. I particularly like the organized answer with the table showing the revised figures - that made life easier during revision.

I am glad that the apparent protein stability problem in the firsts submission turned out to be just a labelling error. It is now shown that the fusion proteins are stable (enough) for the SMLM analysis. The newly added proximity labelling experiments clearly strengthen the paper and confirm the interaction of SctL and SctQ.

We thank the reviewer for this positive evaluation!

The proximity labelling does not reveal interaction of SctQ with YopD, which was tested in vitro by cross-linking. This needs explanation.

The in vitro crosslinking was performed with the effector YopO (in presence of its chaperone SycO), not with YopD. YopO is amongst the enriched effectors presented in Table 1. Nevertheless, SctB/YopD and

4the other translocators are indeed not strongly enriched in the proximity labeling experiment. While it is tempting to speculate whether and why there is a preference for effector binding at the late stages of secretion (which is mostly covered by the experiment), we do not want to overinterpret these findings. However, we now mention this observation explicitly in the legend of Suppl. Table 1, which shows the proximity labeling data.

Although, I appreciate the fact that the authors put a lot of effort in confirming interactions, the *in vitro* crosslinking is rather problematic. Unlike *in vivo* crosslinking, *in vitro* at sufficiently high protein concentrations often protein crosslinks can be achieved that are not found *in vivo*. Since this is just another control here, I do think this experiment can remain, but it needs to be taken with care and is not ultimate proof for interaction.

We fully agree and have now changed the discussion of these results to make this clearer: “Similarly, detecting the interaction of purified SctQ and the effector YopO *in vitro* required crosslinking, which cannot mimic natural conditions (Extended Data Fig. 2).”

I had problems opening the supplemental videos and I wonder whether there is a way to compress them so that they can be published as supplemental material alongside the paper, rather than being on an institutional webpage.

We have now specifically added information on the format of the videos and software that can open these files (e.g. FIJI/ImageJ). With the dataset fully published (and the DOI active), the download should be possible without any restrictions or problems. We would like to leave the raw data, which is the basis for all analysis, unchanged to allow, for example, a re-analysis with different tracking algorithms. However, in response to the reviewer request, we have extracted an exemplary part of one file to visualize the nature of the raw data, and include this qualitative example video as a regular supplemental video.

Final Decision Letter:

Mes 6th November 2023

sag

e: Dear Andreas,

I am pleased to accept your Article "Cytosolic sorting platform complexes shuttle type III secretion system effectors to the injectisome in *Yersinia enterocolitica*" for publication in Nature Microbiology. Thank you for having chosen to submit your work to us and many congratulations.

Over the next few weeks, your paper will be copyedited to ensure that it conforms to Nature Microbiology style. We look particularly carefully at the titles of all papers to ensure that they

5are relatively brief and understandable.

Acceptance of your manuscript is conditional on all authors' agreement with our publication policies (see <https://www.nature.com/nmicrobiol/editorial-policies>). In particular your manuscript must not be published elsewhere and there must be no announcement of the work to any media outlet until the publication date (the day on which it is uploaded onto our website).

Please note that *Nature Microbiology* is a Transformative Journal (TJ). Authors may publish their research with us through the traditional subscription access route or make their paper immediately open access through payment of an article-processing charge (APC). Authors will not be required to make a final decision about access to their article until it has been accepted. [Find out more about Transformative Journals](https://www.springernature.com/gp/open-research/transformative-journals)

Authors may need to take specific actions to achieve [compliance](https://www.springernature.com/gp/open-research/funding/policy-compliance-faqs) with funder and institutional open access mandates. If your research is supported by a funder that requires immediate open access (e.g. according to [Plan S principles](https://www.springernature.com/gp/open-research/plan-s-compliance)) then you should select the gold OA route, and we will direct you to the compliant route where possible. For authors selecting the subscription publication route, the journal's standard licensing terms will need to be accepted, including [self-archiving policies](https://www.nature.com/nature-portfolio/editorial-policies/self-archiving-and-license-to-publish). Those licensing terms will supersede any other terms that the author or any third party may assert apply to any version of the manuscript.

An online order form for reprints of your paper is available at <https://www.nature.com/reprints/author->

reprints.html"><https://www.nature.com/reprints/author-reprints.html>. All co-authors, authors' institutions and authors' funding agencies can order reprints using the form appropriate to their geographical region.

Congratulations once again and I look forward to seeing the article published.

With kind regards,
[redacted]

P.S. Click on the following link if you would like to recommend Nature Microbiology to your librarian <http://www.nature.com/subscriptions/recommend.html#forms>

** Visit the Springer Nature Editorial and Publishing website at www.springernature.com/editorial-and-publishing-jobs for more information about our career opportunities. If you have any questions please click here.**